# Error Analysis of a PFEM Based on the Euler Semi-Implicit Scheme for the Unsteady MHD Equations

**DOI:** 10.3390/e24101395

**Published:** 2022-09-30

**Authors:** Kaiwen Shi, Haiyan Su, Xinlong Feng

**Affiliations:** College of Mathematics and System Sciences, Xinjiang University, Urumqi 830046, China

**Keywords:** MHD equations, PFEM, semi-implicit scheme, error estimates, LBB condition

## Abstract

In this article, we mainly consider a first order penalty finite element method (PFEM) for the 2D/3D unsteady incompressible magnetohydrodynamic (MHD) equations. The penalty method applies a penalty term to relax the constraint “∇·u=0”, which allows us to transform the saddle point problem into two smaller problems to solve. The Euler semi-implicit scheme is based on a first order backward difference formula for time discretization and semi-implicit treatments for nonlinear terms. It is worth mentioning that the error estimates of the fully discrete PFEM are rigorously derived, which depend on the penalty parameter ϵ, the time-step size τ, and the mesh size *h*. Finally, two numerical tests show that our scheme is effective.

## 1. Introduction

The magnetohydrodynamic (MHD) describes the dynamic behavior of a conducting fluid under external electromagnetic field, which is the coupling of the Navier–Stokes (NS) system and Maxwell’s system. It has wide practical applications in geophysics, astrophysics, and confinement for controlled thermonuclear fusion (cf. [1,2,3]). Concerning the corresponding extensive theoretical modeling/numerical analysis of the MHD system, we refer to [1,2,3,4,5,6,7,8,9,10,11,12] and the references therein.

In this paper, we mainly consider the 2D/3D unsteady incompressible MHD equations. This model is a coupled strongly nonlinear system, and it is a saddle point problem due to the incompressible constraint. Therefore, it is necessary to construct unconditionally stable and decoupled algorithms for our model. For the time discretization, it is well-known that simple discretizations, like fully explicit or implicit type schemes, can lead to considerable instabilities or suffer from costly time expense. Recently, Euler semi-implicit schemes for some evolution differential equations have been given in [4,13,14]. This method is unconditionally stable. For the saddle point problem, there are many methods to release the incompressibility constraint for incompressible flow such as the projection method, the pressure stabilization method, the artificial compressibility method, and the penalty method (see also [15,16,17,18,19,20]).

It is worth mentioning that the penalty method is the simplest and the most basic of these methods mentioned above. For the penalty method, it can be traced back to [21]. Then, the optimal error estimate of the unsteady NS system based on the penalty finite element method (PFEM) was given in [22]. A PFEM of a Euler implicit/explicit scheme for the unsteady NS system was proposed in [23]. An error estimate of the unsteady NS system based on the $P_{1}$ nonconforming PFEM was given in [24]. The authors study the PFEM of the steady MHD equations in [25]. A decoupling PFEM for the steady incompressible MHD equations was given in [18].

The aim of this paper is to develop a first order linear and decoupled scheme. We adopt an implicit scheme for the linear terms and semi-implicit treatments for nonlinear terms. Meanwhile, the penalty method is used for fluid equations. This method decouples the MHD equations into two small equations; one is the equations of the velocity and magnetic field (u,B), and the other is the equation of pressure *p*. Then, the error estimates for the developed scheme are present, which depend on the penalty parameter ϵ, the time-step size τ, and the mesh size *h*. Finally, we give two numerical tests to verify the theoretical results of our method.

The structure of the paper is as follows: in Section 2, we present the model and estimates for the solutions of penalty MHD equations. In Section 3, we give the Euler semi-implicit scheme in the case of time discretization and the convergence rate of the semi-discrete solutions. In Section 4, we give the PFEM based on the Euler semi-implicit scheme. In Section 5, we give the error estimates of the fully discrete solutions. In Section 6, the error estimates are obtained for our scheme. In Section 7, two numerical tests show that our scheme is effective. Finally, we give some conclusions.

## 2. Functional Setting of the Unsteady MHD Equation

In this paper, we consider the unsteady incompressible MHD equations as follows:(1)$$\left\{\begin{array}{c} u_{t}-\nu \Delta u+\left(u\cdot \nabla \right)u+\nabla p+SB\times \nabla \times B=f,\;\mathrm{in}\;D_{T},\\ \nabla \cdot u=0,\;\mathrm{in}\;D_{T},\\ B_{t}+\mu \nabla \times \nabla \times B-\nabla \times \left(u\times B\right)=g,\;\mathrm{in}\;D_{T},\\ \nabla \cdot B=0,\;\mathrm{in}\;D_{T},\\ u\left(0\right)=u^{0},\;B\left(0\right)=B^{0},\;\mathrm{in}\;D,\\ u=0,\;B\times n=0,\;\mathrm{on}\;\gamma , \end{array}\right.$$
where $D_{T}=D\times \left[0,T\right]$, Γ=∂D×[0,T], $D\subset R^{d}\left(d=2\mathrm{or}3\right)$ stands for a bounded, convex, and open domain with the boundary ∂D, T is the final time. Here, u,p,B are the velocity, the pressure, and the magnetic field, f is the external force term, g is the known applied current with ∇·g=0, and n denotes the outward normal on ∂D. For the physical parameters, $\nu^{-1}=R_{e}$ (fluid Reynolds number), $\mu^{-1}=R_{m}$ (magnetic Reynolds number) and *S* is the coupling coefficient.

Next, we give a penalty method for the unsteady MHD equations. Instead of solving (Equation 1), we solve $\left(u_{\epsilon },p_{\epsilon },B_{\epsilon }\right)$ from the penalty MHD equations:(2)$$\left\{\begin{array}{c} u_{\epsilon t}-\nu \Delta u_{\epsilon }+\tilde{b}\left(u_{\epsilon },u_{\epsilon }\right)+\nabla p_{\epsilon }+SB_{\epsilon }\times \nabla \times B_{\epsilon }=f,\;\mathrm{in}\;D_{T},\\ \nabla \cdot u_{\epsilon }+\frac{\epsilon }{\nu }p_{\epsilon }=0,\;\mathrm{in}\;D_{T},\\ B_{\epsilon t}+\mu \nabla \times \nabla \times B_{\epsilon }-\nabla \times \left(u_{\epsilon }\times B_{\epsilon }\right)=g,\;\mathrm{in}\;D_{T},\\ \nabla \cdot B_{\epsilon }=0,\;\mathrm{in}\;D_{T},\\ u_{\epsilon }\left(0\right)=u^{0},\;B_{\epsilon }\left(0\right)=B^{0},\;\mathrm{in}\;D,\\ u_{\epsilon }=0,\;B_{\epsilon }\times n=0,\;\mathrm{on}\;\gamma , \end{array}\right.$$
where 0<ϵ<1 is the penalty parameter; $\tilde{b}\left(u,v\right)=\left(u\cdot \nabla \right)v+\frac{1}{2}\left(\nabla \cdot u\right)v$ is the modified nonlinear term.

Then, we give some notations and estimates for MHD equations. For 1≤r≤∞, $L^{r}\left(D\right)$ denotes the usual Lebesgue space on D with the norm ${∥\cdot ∥}_{L^{r}}$. The inner product of the space $L^{2}\left(D\right)$ is denoted by (·,·) that is $\left(u,v\right)=\int_{D}uvdx$, and the norm of the space $L^{2}\left(D\right)$ is denoted by ∥·∥. For all non-negative integers *k* and *r*, $W^{k,r}\left(D\right)$ stands for the standard Sobolev space equipped with the standard Sobolev norm ${∥\cdot ∥}_{k,r}$. The norm of the space $W^{k,2}\left(D\right)$ is represented by ${∥\cdot ∥}_{k}$. The functions and spaces of vectors are represented in boldface.

Next, we give several function spaces
$$\begin{array}{cc} & \tilde{H}:=\left\{v\in L^{2}\left(D\right):\nabla \cdot v=0,v\cdot n{|}_{\partial D}=0\right\},\\ & X:=H_{0}^{1}\left(D\right)=\left\{v\in H^{1}{\left(D\right)}^{d}:v{|}_{\partial D}=0\right\},\\ & M:=L_{0}^{2}\left(D\right)=\left\{q\in L^{2}\left(D\right):\int_{D}qdx=0\right\},\\ & W:=H_{n}^{1}\left(D\right)=\left\{w\in H^{1}{\left(D\right)}^{d}:w\cdot n{|}_{\partial D}=0\right\},\\ & X_{0}:=\left\{v\in X:\nabla \cdot v=0\right\},\\ & W_{0}:=\left\{w\in W:\nabla \cdot w=0\right\}. \end{array}$$

We define $A_{1}u=-\Delta u$ and $A_{1\epsilon }u=-\Delta u-\frac{1}{\epsilon }\nabla \nabla \cdot u$, which are the operators associated with NS equations and the penalty NS equations. They are the positive self-adjoint operators from $D\left(A_{1}\right)=H^{2}\left(D\right)\cap X$ onto $L^{2}\left(D\right)$ and the powers $A_{1}^{\alpha }$ and $A_{1\epsilon }^{\alpha }$(α∈R) are well defined. Similarly, we define the operator $A_{2}=P_{H}\left(\nabla \times \nabla \times -\nabla \nabla \cdot \right):D\left(A_{2}\right)\to \tilde{H}$, where $D\left(A_{2}\right)=H^{2}\left(D\right)\cap W$ and $P_{H}$ is the $L^{2}$-orthogonal projector (cf. [7,26,27]). Thus, we have
$$\begin{array}{c} \left(A_{1}u,v\right)=\left(A_{1}^{\frac{1}{2}}u,A_{1}^{\frac{1}{2}}v\right)=\left(\nabla u,\nabla v\right),\;\forall u,v\in X,\\ \left(A_{1\epsilon }u,v\right)=\left(A_{1\epsilon }^{\frac{1}{2}}u,A_{1\epsilon }^{\frac{1}{2}}v\right)=\left(\nabla u,\nabla v\right)+\frac{1}{\epsilon }\left(\nabla \cdot u,\nabla \cdot v\right),\;\forall u,v\in X,\\ \left(A_{2}B,C\right)=\left(A_{2}^{\frac{1}{2}}B,A_{2}^{\frac{1}{2}}C\right)=\left(\nabla \times B,\nabla \times C\right)+\left(\nabla \cdot B,\nabla \cdot C\right),\;\forall B,C\in W. \end{array}$$Define
$$\begin{array}{cc} b(u,v,w) & =\left(u\cdot \nabla v,w\right)+\frac{1}{2}\left(\left(\nabla \cdot u\right)v,w\right)=\frac{1}{2}b\left(u,v,w\right)-\frac{1}{2}b\left(u,w,v\right),\;\forall u,v,w\in X. \end{array}$$Therefore, the trilinear form b(·,·,·) satisfies
(3)$$\begin{array}{c} b(u,v,v)=0,\;\forall u,v\in X. \end{array}$$

A weak formulation for (Equation 1) is as follows: find $\left(u,p,B\right)\in L^{2}\left(0,T;X\right)\times L^{2}\left(0,T;M\right)\times L^{2}\left(0,T;W\right)$ such that, for all (v,q,C)∈X×M×W (cf. [1,4]),
(4)$$\left\{\begin{array}{c} \left(u_{t},v\right)+\nu \left(\nabla u,\nabla v\right)+b\left(u,u,v\right)-\left(\nabla \cdot v,p\right)+\left(\nabla \cdot u,q\right)+S\left(B\times \nabla \times B,v\right)=\left(f,v\right),\\ \left(B_{t},C\right)+\mu \left(\nabla \times B,\nabla \times C\right)-\left(u\times B,\nabla \times C\right)=\left(g,C\right),\\ u\left(0\right)=u^{0},\;B\left(0\right)=B^{0}, \end{array}\right.$$
where $u_{t}\in L^{4/d}\left(0,T;X^{\prime }\right)$, $B_{t}\in L^{4/d}\left(0,T;W^{\prime }\right)$ ($X^{\prime }$ and $W^{\prime }$ are the dual spaces of X and W, respectively), $\nabla \cdot u^{0}=\nabla \cdot B^{0}=0$ and the weak formulation for (Equation 2) is as follows: find $\left(u_{\epsilon },p_{\epsilon },B_{\epsilon }\right)\in L^{2}\left(0,T;X\right)\times L^{2}\left(0,T;M\right)\times L^{2}\left(0,T;W\right)$ such that, for all (v,q,C)∈X×M×W
(5)$$\left\{\begin{array}{c} \left(u_{\epsilon t},v\right)-\nu \left(\nabla u_{\epsilon },\nabla v\right)+b\left(u_{\epsilon },u_{\epsilon },v\right)-\left(\nabla \cdot v,p_{\epsilon }\right)+\left(\nabla \cdot u_{\epsilon },q\right)+S\left(B_{\epsilon }\times \nabla \times B_{\epsilon },v\right)\\ \;+\frac{\epsilon }{\nu }\left(p_{\epsilon },q\right)=\left(f,v\right),\\ \left(B_{\epsilon t},C\right)+\mu \left(\nabla \times B_{\epsilon },\nabla \times C\right)-\left(u_{\epsilon }\times B_{\epsilon },\nabla \times C\right)=\left(g,C\right),\\ u_{\epsilon }\left(0\right)=u^{0},\;B_{\epsilon }\left(0\right)=B^{0}, \end{array}\right.$$
where $u_{\epsilon t}\in L^{4/d}\left(0,T;X^{\prime }\right)$, $B_{\epsilon t}\in L^{4/d}\left(0,T;W^{\prime }\right)$.

**Remark** **1.**
*Taking the $L^{2}$ inner product of the first equation in (Equation 1) with v, the second equation in (Equation 1) with q and summing up the two relations, we obtain the first equation of (Equation 4). Taking the $L^{2}$ inner product of the third equation in (Equation 1) with C, we obtain the second equation of (Equation 4). Equation (Equation 5) can be obtained similarly.*


**Remark** **2.**
*For $\phi \in H_{0}^{2}\left(D\right)$, taking C=∇ϕ in the second equation of (Equation 4), we easily obtain ${\left(\nabla \cdot B\right)}_{t}=0$, which implies $\nabla \cdot B\left(t\right)=\nabla \cdot B^{0}=0$. In addition, the second equation of (Equation 4) has an equivalent form (cf. [1,7])*

$$\begin{array}{c} \left(B_{t},C\right)+\mu \left(\nabla \times B,\nabla \times C\right)+\mu \left(\nabla \cdot B,\nabla \cdot C\right)-\left(u\times B,\nabla \times C\right)=\left(g,C\right), \end{array}$$

*in which ∇·B is used as a penalty term. For ∀t∈(0,T), we choose C=∇ϕ(t)∈W in the second equation of (Equation 4); here, ϕ(t) is generated by the boundary value problem*

$$\left\{\begin{array}{c} \Delta \phi (t)=\nabla \cdot B(t),\\ \frac{\partial \phi }{\partial n}{|}_{\partial D}=0. \end{array}\right.$$

*We can obtain $\frac{1}{2}{∥\nabla \phi \left(t\right)∥}^{2}+\mu \int_{0}^{T}{∥\nabla \cdot B\left(t\right)∥}^{2}dt=0$, which also implies ∇·B(t)=0.*


Using the operators $A_{1\epsilon },A_{2}$, we can rewrite the penalized system (Equation 5) as
(6)$$\left\{\begin{array}{c} u_{\epsilon t}+\nu A_{1\epsilon }u_{\epsilon }+\tilde{b}\left(u_{\epsilon },u_{\epsilon }\right)+S\left(B_{\epsilon }\times \nabla \times B_{\epsilon }\right)=f,\\ B_{\epsilon t}+\mu A_{2}B_{\epsilon }-\nabla \times \left(u_{\epsilon }\times B_{\epsilon }\right)=g. \end{array}\right.$$

Referring to [2,6,13,28], the following estimates hold: (7)$$\begin{array}{ccc} & & {∥v∥}_{L^{p}}\leq c{∥v∥}_{1},\;2\leq p\leq 6,\;v\in H^{1}\left(D\right), \end{array}$$(8)$$\begin{array}{ccc} & & {∥v∥}_{L^{\infty }}+{∥\nabla v∥}_{L^{3}}\leq {c∥v∥}_{1}^{\frac{1}{2}}{∥v∥}_{2}^{\frac{1}{2}},\;v\in H^{2}\left(D\right), \end{array}$$(9)$$\begin{array}{ccc} & & {∥C∥}_{1}\leq c∥\nabla \times C∥+c∥\nabla \cdot C∥,\;\forall C\in W, \end{array}$$(10)$$\begin{array}{ccc} & & (B\times \nabla \times C,v)=(v\times B,\nabla \times C),\;\forall B,C\in W,\;v\in X, \end{array}$$(11)$$\begin{array}{ccc} & & ∥\nabla \times C∥\leq \sqrt{2}∥\nabla C∥,\;∥\nabla \cdot C∥\leq \sqrt{d}∥\nabla C∥,\;\forall C\in W, \end{array}$$(12)$$\begin{array}{ccc} & & \nabla \times \left(v\times C\right)=v\left(\nabla \cdot C\right)-C\left(\nabla \cdot v\right)+\left(C\cdot \nabla \right)v-\left(v\cdot \nabla \right)C,\;\forall v,C\in H^{1}\left(D\right), \end{array}$$
where c(D)>0 is a constant, which have different values in different cases.

In this paper, C>0 denotes a constant depending on $\left(\nu ,\mu ,S,D,T,u_{0},B_{0},f,g\right)$, which may have different values in different cases. We make the following assumptions for (Equation 1), which specify the regularity of the data and the smoothness of the domain D (cf. [1,4]).

**Assumption** **1.**
*The initial data $u_{0}\in X_{0}\cap H^{2}\left(\Omega \right)$ and $B_{0}\in W_{0}\cap H^{2}\left(\Omega \right),$ the external force f, and the applied current g satisfy the following bound:*

$$∥u_{0}{∥}_{2}+{∥B_{0}∥}_{2}+\underset{0\leq t\leq T}{\sup}\left\{∥f\left(t\right)∥+∥g\left(t\right)∥+∥f_{t}\left(t\right)∥+∥g_{t}\left(t\right)∥\right\}\leq C.$$



Assumption 1 ensures that there is a unique strong solution for (Equation 1) over some time interval [0,T); we have (cf. [5])
$$\begin{array}{c} \begin{array}{c} u\in C\left(0,T;X\right)\cap L^{2}\left(0,T;H^{2}\left(D\right)\right),\;p\in L^{2}\left(0,T;H^{1}\left(D\right)\cap M\right),\\ B\in C\left(0,T;W\right)\cap L^{2}\left(0,T;H^{2}\left(D\right)\right), \end{array} \end{array}$$
such that $u_{t},B_{t}\in L^{2}\left(0,T;L^{2}\left(D\right)\right)$, and Equation (Equation 4) holds for almost all t∈[0,T). If the data $u_{0},B_{0},f$ and g are sufficiently small, then the solution exists for any T>0 and satisfies
(13)$$\begin{array}{c} \underset{0\leq t\leq T}{\sup}{(∥u\left(t\right)∥}_{1}+{∥B\left(t\right)∥}_{1})<\infty . \end{array}$$

**Assumption** **2.**
*The problem (Equation 4) has a weak solution (u(t),p(t),B(t)) satisfying $u\in L^{2}\left(0,T;X_{0}\right)$, $p\in L^{2}\left(0,T;M\right)$ and $B\in L^{2}\left(0,T;W_{0}\right)$ such that*

$$\begin{array}{c} \int_{0}^{T}{(∥\nabla u\left(t\right)∥}^{4}{+∥\nabla \times B\left(t\right)∥}^{4})dt\leq C. \end{array}$$



**Remark** **3.**
*Instead of assuming the data are small or strong condition (Equation 13) holds, we give the Assumption 2 to guarantee the uniqueness of weak solution to the 3D MHD problem on interval [0,T] (see [4,5,28]).*


**Assumption** **3.**
*Assume that the boundary of D is smooth so that the unique solution (v,q)∈X×M of the steady Stokes problem*

$$\begin{array}{c} {-\Delta v+\nabla q=f,\;\nabla \cdot v=0,\;\mathrm{in}\;D,\;v|}_{\partial D}=0, \end{array}$$

*for prescribed $f\in L^{2}\left(D\right)$ satisfies*

$$\begin{array}{c} {∥v∥}_{2}+{∥q∥}_{1}\leq c∥f∥; \end{array}$$

*and Maxwell’s equations*

$$\begin{array}{c} {\nabla \times \nabla \times C=g,\;\nabla \cdot C=0,\;\mathrm{in}\;D,\;C\times n|}_{\partial D}=0, \end{array}$$

*for the prescribed $g\in L^{2}\left(D\right)$ admit a unique solution $C\in W_{0}$, which satisfies*

$$\begin{array}{c} {∥C∥}_{2}\leq c∥g∥. \end{array}$$



**Remark** **4.**
*The validity of Assumption 3 is known if ∂D is of $C^{2}$, or if D is a convex polyhedron (see [4]).*

*Next, we need the following lemma given in [26].*


**Lemma** **1.**
*There exists a constant $c_{1}>0$ depending only on D and such that, for sufficiently small ϵ, we have*

$$\begin{array}{ccc} & & ∥\Delta u∥\leq c_{1}∥A_{1\epsilon }u∥,\;\forall u\in H^{2}\left(D\right)\cap X,\\ & & ∥\nabla u∥\leq c_{1}∥A_{1\epsilon }^{\frac{1}{2}}u∥,\;\forall u\in X,\\ & & ∥A_{1\epsilon }^{-1}u∥\leq c_{1}{∥u∥}_{-2},\;\forall u\in H^{-2}\left(D\right), \end{array}$$

*where $H^{-2}\left(D\right)$ is the dual space of $H^{2}\left(D\right)\cap X$, and ${∥\cdot ∥}_{-2}$ is the corresponding norm.*


**Theorem** **1.**
*Under Assumptions 1–3, the solution (u(t),p(t),B(t)) of the problem (Equation 4) satisfies the estimates*

$$\begin{array}{ccc} & & \underset{0\leq t\leq T}{\sup}\{∥u_{t}{\left(t\right)∥}^{2}+∥B_{t}{\left(t\right)∥}^{2}+{∥u\left(t\right)∥}_{2}^{2}+{∥p\left(t\right)∥}_{1}^{2}+{∥B\left(t\right)∥}_{2}^{2}\}\\ & & \;+\int_{0}^{T}(∥u_{tt}{∥}_{X_{0}^{\prime }}^{2}+∥B_{tt}{∥}_{W^{\prime }}^{2}+∥\nabla u_{t}{∥}^{2}+∥\nabla B_{t}{∥}^{2})dt\leq C,\\ & & \underset{0\leq t\leq T}{\sup}\{\sigma \left(t\right)∥u_{t}{\left(t\right)∥}_{1}^{2}+\sigma \left(t\right)∥B_{t}{\left(t\right)∥}_{1}^{2}\}+\int_{0}^{T}\sigma \left(t\right)(∥u_{tt}{∥}^{2}+∥B_{tt}{∥}^{2})dt\\ & & \;+\int_{0}^{T}\sigma \left(t\right)(∥u_{t}{∥}_{2}^{2}+∥B_{t}{∥}_{2}^{2}+∥p_{t}{∥}_{1}^{2})dt\leq C, \end{array}$$

*where*

$$\begin{array}{c} ∥u_{tt}{∥}_{X_{0}^{\prime }}=\underset{v\in X_{0}}{\sup}\frac{\left(u_{tt},v\right)}{∥\nabla v∥},\;{∥B_{tt}∥}_{W^{\prime }}=\underset{C\in W}{\sup}\frac{\left(B_{tt},C\right)}{∥\nabla C∥}. \end{array}$$



For the proof of these results, we can refer to [1,4].

**Theorem** **2.**
*Under Assumptions 1–3, the solution $\left(u_{\epsilon }\left(t\right),p_{\epsilon }\left(t\right),B_{\epsilon }\left(t\right)\right)$ of the problem (Equation 3) satisfies the estimates*

$$\begin{array}{ccc} & & \underset{0\leq t\leq T}{\sup}\{∥u_{\epsilon t}{\left(t\right)∥}^{2}+∥B_{\epsilon t}{\left(t\right)∥}^{2}+∥u_{\epsilon }{\left(t\right)∥}_{2}^{2}+∥p_{\epsilon }{\left(t\right)∥}_{1}^{2}+∥B_{\epsilon }\left(t\right){∥}_{2}^{2}\}\\ & & \;+\int_{0}^{T}(∥A_{1\epsilon }^{-\frac{1}{2}}u_{\epsilon tt}{∥}^{2}+∥A_{2}^{-\frac{1}{2}}B_{\epsilon tt}{∥}^{2}+∥\nabla u_{\epsilon t}{∥}^{2}+∥\nabla B_{\epsilon t}{∥}^{2})dt\leq C,\\ & & \underset{0\leq t\leq T}{\sup}\{\sigma \left(t\right)∥u_{\epsilon t}{\left(t\right)∥}_{1}^{2}+\sigma \left(t\right)∥B_{\epsilon t}{\left(t\right)∥}_{1}^{2}\}+\int_{0}^{T}\sigma \left(t\right)(∥u_{\epsilon tt}{∥}^{2}+∥B_{\epsilon tt}{∥}^{2}+∥p_{\epsilon t}{∥}^{2})dt\leq C. \end{array}$$



We can finish the proof by a similar technique used in the proof of Theorem 1.

**Theorem** **3.**
*Under Assumptions 1–3, we have the following estimate (cf. [26,29])*

$$\begin{array}{ccc} & & \underset{0\leq t\leq T}{\sup}{\{\sigma \left(t\right)}^{\frac{1}{2}}(∥u\left(t\right)-u_{\epsilon }\left(t\right)∥+∥B\left(t\right)-B_{\epsilon }\left(t\right)∥)+\sigma \left(t\right)(∥u\left(t\right)-u_{\epsilon }{\left(t\right)∥}_{1}+∥B\left(t\right)-B_{\epsilon }{\left(t\right)∥}_{1})\}\\ & & \;+{\left(\int_{0}^{T}\sigma^{2}\left(t\right){∥p\left(t\right)-p_{\epsilon }\left(t\right)∥}^{2}dt\right)}^{\frac{1}{2}}\leq C\epsilon . \end{array}$$



**Proof.** We first consider the linear form of MHD equations. Then, we subtract the penalized linear MHD equations from the linear MHD equations to obtain their error equations. The error estimate in linear form is obtained through its dual problem
$$\begin{array}{c} \int_{0}^{T}(∥u\left(t\right)-u_{\epsilon }\left(t\right)∥+∥B\left(t\right)-B_{\epsilon }\left(t\right)∥)dt\leq C\epsilon . \end{array}$$Next, we obtain the following error estimate by choosing an appropriate $L^{2}$ inner product for the error equations
$$\begin{array}{cc} & \underset{0\leq t\leq T}{\sup}{\{\sigma \left(t\right)}^{\frac{1}{2}}(∥u\left(t\right)-u_{\epsilon }\left(t\right)∥+∥B\left(t\right)-B_{\epsilon }\left(t\right)∥)+\sigma \left(t\right)(∥u\left(t\right)-u_{\epsilon }{\left(t\right)∥}_{1}+∥B\left(t\right)-B_{\epsilon }{\left(t\right)∥}_{1})\}\\ & \;+{\left(\int_{0}^{T}\sigma^{2}\left(t\right){∥p\left(t\right)-p_{\epsilon }\left(t\right)∥}^{2}dt\right)}^{\frac{1}{2}}\leq C\epsilon . \end{array}$$Finally, we transform the nonlinear MHD equations into an intermediate linear equations, and then obtain Theorem 3 by applying a suitable $L^{2}$ inner product to this system and using the previous result. □

In addition, we need the following discrete Gronwall lemma (cf. [4,30,31]).

**Lemma** **2.**
*Let $a_{n},b_{n},d_{n}$ and $C_{0}$ be nonnegative numbers for integer n≥0, such that*

(14)
$$\begin{array}{c} a_{m}+\tau \sum_{n=1}^{m}b_{n}\leq \tau \sum_{n=0}^{m-1}d_{n}a_{n}+C_{0},\;\forall m\geq 1. \end{array}$$

*Then*

(15)
$$\begin{array}{c} a_{m}+\tau \sum_{n=1}^{m}b_{n}\leq C_{0}\exp\left(\tau \sum_{n=0}^{m-1}d_{n}\right),\;\forall m\geq 1. \end{array}$$



## 3. The Euler Semi-Implicit Scheme and Its Error Estimates: Time Discretization

In this section, we consider a time discretization for the penalty MHD system (Equation 6). Let $\tau =\frac{T}{N}$ be the time-step size and N>0 is an integer. Then, $t_{n}=n\tau$, n=1,2,⋯,N denote the discrete time levels. The time-discrete approximations to $\left(u_{\epsilon }\left(t_{n}\right),p_{\epsilon }\left(t_{n}\right),B_{\epsilon }\left(t_{n}\right)\right)$ will be denoted by $\left(u_{\epsilon }^{n},p_{\epsilon }^{n},B_{\epsilon }^{n}\right)$ for all 1≤n≤N. Consider the Euler semi-implicit time-stepping algorithm: Given $(u_{\epsilon }^{n-1},B_{\epsilon }^{n-1})\in X\times W$, find $(u_{\epsilon }^{n},B_{\epsilon }^{n})\in X\times W$, $p_{\epsilon }^{n}\in M$ such that
(16)$$\left\{\begin{array}{c} \left(d_{t}u_{\epsilon }^{n},v\right)-\nu \left(\nabla u_{\epsilon }^{n},\nabla v\right)+b\left(u_{\epsilon }^{n-1},u_{\epsilon }^{n},v\right)-\left(\nabla \cdot v,p_{\epsilon }^{n}\right)+\left(\nabla \cdot u_{\epsilon }^{n},q\right)\\ +S\left(B_{\epsilon }^{n-1}\times \nabla \times B_{\epsilon }^{n},v\right)+\frac{\epsilon }{\nu }\left(p_{\epsilon }^{n},q\right)=\left(f\left(t_{n}\right),v\right),\\ \left(d_{t}B_{\epsilon }^{n},C\right)+\mu \left(\nabla \times B_{\epsilon }^{n},\nabla \times C\right)-\left(u_{\epsilon }^{n}\times B_{\epsilon }^{n-1},\nabla \times C\right)=\left(g\left(t_{n}\right),C\right), \end{array}\right.$$
where $\left(u_{\epsilon }^{0},B_{\epsilon }^{0}\right)=\left(u^{0},B^{0}\right)$, $d_{t}u^{n}=\frac{1}{\tau }\left(u^{n}-u^{n-1}\right)$. We can rewrite (Equation 16) as
(17)$$\left\{\begin{array}{c} d_{t}u_{\epsilon }^{n}+\nu A_{1\epsilon }u_{\epsilon }^{n}+\tilde{b}\left(u_{\epsilon }^{n-1},u_{\epsilon }^{n}\right)+S\left(B_{\epsilon }^{n-1}\times \nabla \times B_{\epsilon }^{n}\right)=f^{n},\\ d_{t}B_{\epsilon }^{n}+\mu A_{2}B_{\epsilon }^{n}-\nabla \times \left(u_{\epsilon }^{n}\times B_{\epsilon }^{n-1}\right)=g^{n}. \end{array}\right.$$

**Remark** **5.**
*Since $\nabla \cdot B^{0}=0$, we take C=∇ϕ with $\phi \in H_{0}^{2}\left(D\right)$ and deduce from the second equation in (Equation 16) and the identity ∇×∇ϕ=0 that $\nabla \cdot B^{n}=0$ for all 0≤n≤N.*


Next, we give a priori bound of the scheme (Equation 17).

**Theorem** **4.**
*Under Assumptions 1–3, we have a priori bound*

$$\begin{array}{c} ∥u_{\epsilon }^{m}{∥}^{2}+S∥B_{\epsilon }^{m}{∥}^{2}+\tau \sum_{n=1}^{m}(\nu ∥A_{1\epsilon }^{\frac{1}{2}}u_{\epsilon }^{n}{∥}^{2}+S\mu ∥A_{2}^{\frac{1}{2}}B_{\epsilon }^{n}{∥}^{2})+\tau \sum_{n=1}^{m}(∥d_{t}u_{\epsilon }^{n}{∥}^{2}+S∥d_{t}B_{\epsilon }^{n}{∥}^{2})\tau \leq C, \end{array}$$

*for all 1≤m≤N.*


**Proof.** Taking the $L^{2}$ inner product of the first equation in (Equation 17) with $2u_{\epsilon }^{n}\tau$, and the second equation in (Equation 17) with $2SB_{\epsilon }^{n}\tau$, we obtain
(18)$$\begin{array}{c} ∥u_{\epsilon }^{n}{∥}^{2}+S∥B_{\epsilon }^{n}{∥}^{2}-(∥u_{\epsilon }^{n-1}{∥}^{2}+S∥B_{\epsilon }^{n-1}{∥}^{2})+(∥d_{t}u_{\epsilon }^{n}{∥}^{2}+S∥d_{t}B_{\epsilon }^{n}{∥}^{2})\tau^{2}\\ \;+2\nu ∥A_{1\epsilon }^{\frac{1}{2}}u_{\epsilon }^{n}{∥}^{2}\tau +2S\mu ∥A_{2}^{\frac{1}{2}}B_{\epsilon }^{n}{∥}^{2}\tau \leq 2\left(f^{n},u_{\epsilon }^{n}\right)\tau +2S\left(g^{n},B_{\epsilon }^{n}\right)\tau . \end{array}$$By using (Equation 7), (Equation 9), Lemma 1 and the Young inequality, we have
$$\begin{array}{c} 2|\left(f^{n},u_{\epsilon }^{n}\right)|\leq 2∥f^{n}∥\;∥u_{\epsilon }^{n}∥\leq \nu ∥A_{1\epsilon }^{\frac{1}{2}}u_{\epsilon }^{n}{∥}^{2}+\frac{c}{\tau }\int_{t_{n-1}}^{t_{n}}{∥f\left(t\right)∥}^{2}dt,\\ 2S|\left(g^{n},B_{\epsilon }^{n}\right)|\leq 2S∥g^{n}∥\;∥B_{\epsilon }^{n}∥\leq S\mu ∥A_{2}^{\frac{1}{2}}B_{\epsilon }^{n}{∥}^{2}+\frac{c}{\tau }\int_{t_{n-1}}^{t_{n}}{∥g\left(t\right)∥}^{2}dt. \end{array}$$Combining the above inequalities with (Equation 18), we obtain
$$\begin{array}{c} ∥u_{\epsilon }^{n}{∥}^{2}+S∥B_{\epsilon }^{n}{∥}^{2}-(∥u_{\epsilon }^{n-1}{∥}^{2}+S∥B_{\epsilon }^{n-1}{∥}^{2})+(∥d_{t}u_{\epsilon }^{n}{∥}^{2}+S∥d_{t}B_{\epsilon }^{n}{∥}^{2})\tau^{2}\\ \;+\nu ∥A_{1\epsilon }^{\frac{1}{2}}u_{\epsilon }^{n}{∥}^{2}\tau +S\mu ∥A_{2}^{\frac{1}{2}}B_{\epsilon }^{n}{∥}^{2}\tau \leq c\int_{t_{n-1}}^{t_{n}}{(∥f\left(t\right)∥}^{2}+{∥g\left(t\right)∥}^{2})dt. \end{array}$$Summing the above inequality from 1 to *m*, we derive
(19)$$\begin{array}{ccc} & & ∥u_{\epsilon }^{m}{∥}^{2}+S∥B_{\epsilon }^{m}{∥}^{2}+\tau \sum_{n=1}^{m}(∥d_{t}u_{\epsilon }^{n}{∥}^{2}+S∥d_{t}B_{\epsilon }^{n}{∥}^{2})\tau +\tau \sum_{n=1}^{m}(\nu ∥A_{1\epsilon }^{\frac{1}{2}}u_{\epsilon }^{n}{∥}^{2}\tau +S\mu ∥A_{2}^{\frac{1}{2}}B_{\epsilon }^{n}{∥}^{2})\\ & & \;\leq ∥u_{\epsilon }^{0}{∥}^{2}+S∥B_{\epsilon }^{0}{∥}^{2}+c\int_{0}^{T}{(∥f\left(t\right)∥}^{2}+{∥g\left(t\right)∥}^{2})dt, \end{array}$$
for all 1≤m≤N. Using Assumption 1 and (Equation 19), we obtain Theorem 4. □

Next, we establish the error estimates in time for the Euler semi-implicit scheme (Equation 17). To do this, subtracting (Equation 17) from (Equation 6) and setting $e_{u}^{n}=u_{\epsilon }\left(t_{n}\right)-u_{\epsilon }^{n}$, $e_{B}^{n}=B_{\epsilon }\left(t_{n}\right)-B_{\epsilon }^{n}$, we have
(20)$$\begin{array}{ccc} & & d_{t}e_{u}^{n}+\nu A_{1\epsilon }e_{u}^{n}+\tilde{b}\left(e_{u}^{n-1},u_{\epsilon }\left(t_{n}\right)\right)+\tilde{b}\left(u_{\epsilon }^{n-1},e_{u}^{n}\right)+Se_{B}^{n-1}\times \nabla \times B_{\epsilon }\left(t_{n}\right)\\ & & \;+SB_{\epsilon }^{n-1}\times \nabla \times e_{B}^{n}=R_{1}^{n}, \end{array}$$
(21)$$\begin{array}{cc} & d_{t}e_{B}^{n}+\mu A_{2}e_{B}^{n}-\nabla \times \left(e_{u}^{n}\times B_{\epsilon }\left(t_{n-1}\right)\right)-\nabla \times \left(u_{\epsilon }^{n}\times e_{B}^{n-1}\right)=R_{2}^{n}, \end{array}$$
where
(22)$$\begin{array}{cc} R_{1}^{n}= & -\frac{1}{\tau }\int_{t_{n-1}}^{t_{n}}\left(t-t_{n-1}\right)u_{\epsilon tt}\left(t\right)dt+\tilde{b}\left(u_{\epsilon }\left(t_{n-1}\right)-u_{\epsilon }\left(t_{n}\right),u_{\epsilon }\left(t_{n}\right)\right)\\ & +S(\left(B_{\epsilon }\left(t_{n-1}\right)-B_{\epsilon }\left(t_{n}\right)\right)\times \nabla \times B_{\epsilon }\left(t_{n}\right)), \end{array}$$
(23)$$\begin{array}{cc} R_{2}^{n}= & -\frac{1}{\tau }\int_{t_{n-1}}^{t_{n}}\left(t-t_{n-1}\right)B_{\epsilon tt}\left(t\right)dt-\nabla \times \left(u_{\epsilon }\left(t_{n}\right)\times \left(B_{\epsilon }\left(t_{n-1}\right)-B_{\epsilon }\left(t_{n}\right)\right)\right). \end{array}$$

We are now in a position to state and prove two error estimates for the Euler semi-implicit scheme (Equation 17).

**Theorem** **5.**
*Under Assumptions 1–3, we obtain*

$$\begin{array}{c} \underset{1\leq n\leq N}{\sup}(∥u_{\epsilon }\left(t_{n}\right)-u_{\epsilon }^{n}∥+\sqrt{S}∥B_{\epsilon }\left(t_{n}\right)-B_{\epsilon }^{n}∥)+\tau^{\frac{1}{2}}\sum_{n=1}^{N}(\sqrt{\nu }∥u_{\epsilon }\left(t_{n}\right)-u_{\epsilon }^{n}{∥}_{1}+\sqrt{S\mu }∥B_{\epsilon }\left(t_{n}\right)-B_{\epsilon }^{n}{∥}_{1})\leq C\tau . \end{array}$$



**Proof.** Taking the inner $L^{2}$ product of (Equation 20) with $2e_{u}^{n}\tau$, (Equation 21) with $2e_{B}^{n}\tau$, thanks to (Equation 4) and (Equation 10), we deduce that
(24)$$\begin{array}{cc} & ∥e_{u}^{n}{∥}^{2}+S∥e_{B}^{n}{∥}^{2}-(∥e_{u}^{n-1}{∥}^{2}+S∥e_{B}^{n-1}{∥}^{2})+2\nu ∥A_{1\epsilon }^{\frac{1}{2}}e_{u}^{n}{∥}^{2}\tau +2S\mu ∥A_{2}^{\frac{1}{2}}e_{B}^{n}{∥}^{2}\tau +2b\left(e_{u}^{n-1},u_{\epsilon }\left(t_{n}\right),e_{u}^{n}\right)\tau \\ & \;+2S\left(e_{B}^{n-1}\times \nabla \times B_{\epsilon }\left(t_{n}\right),e_{u}^{n}\right)\tau -2S\left(u_{\epsilon }\left(t_{n}\right)\times e_{B}^{n-1},\nabla \times e_{B}^{n}\right)\tau \leq 2\left(R_{1}^{n},e_{u}^{n}\right)\tau +2\left(R_{2}^{n},e_{B}^{n}\right)\tau . \end{array}$$By using (Equation 7)–(Equation 11) and Lemma 1, we obtain
$$\begin{array}{cc} 2|b(e_{u}^{n-1},u_{\epsilon }\left(t_{n}\right),e_{u}^{n})| & \leq c∥e_{u}^{n-1}∥\;∥e_{u}^{n}{∥}_{1}∥u_{\epsilon }\left(t_{n}\right){∥}_{2}\leq \frac{\nu }{6}∥A_{1\epsilon }^{\frac{1}{2}}e_{u}^{n}{∥}^{2}+c∥u_{\epsilon }\left(t_{n}\right){∥}_{2}^{2}{∥e_{u}^{n-1}∥}^{2},\\ 2S|(e_{B}^{n-1}\times \nabla \times B_{\epsilon }\left(t_{n}\right),e_{u}^{n})| & \leq c∥e_{B}^{n-1}∥\;∥e_{u}^{n}{∥}_{1}∥B_{\epsilon }\left(t_{n}\right){∥}_{2}\leq \frac{\nu }{6}∥A_{1\epsilon }^{\frac{1}{2}}e_{u}^{n}{∥}^{2}+c∥B_{\epsilon }\left(t_{n}\right){∥}_{2}^{2}{∥e_{B}^{n-1}∥}^{2},\\ 2S|(u_{\epsilon }\left(t_{n}\right)\times e_{B}^{n-1},\nabla \times e_{B}^{n})| & \leq c∥u_{\epsilon }\left(t_{n}\right){∥}_{2}∥e_{B}^{n}{∥}_{1}∥e_{B}^{n-1}∥\leq \frac{S\mu }{4}∥A_{2}^{\frac{1}{2}}e_{B}^{n}{∥}^{2}+c∥u_{\epsilon }\left(t_{n}\right){∥}_{2}^{2}{∥e_{B}^{n-1}∥}^{2}. \end{array}$$Similarly, we can derive
$$\begin{array}{cc} 2|(R_{1}^{n},e_{u}^{n})|\leq & \frac{\nu }{6}∥A_{1\epsilon }^{\frac{1}{2}}e_{u}^{n}{∥}^{2}+c\tau \int_{t_{n-1}}^{t_{n}}∥A_{1\epsilon }^{-\frac{1}{2}}u_{\epsilon tt}{∥}^{2}dt+c\tau ∥u_{\epsilon }\left(t_{n}\right){∥}_{1}^{2}\int_{t_{n-1}}^{t_{n}}{∥u_{\epsilon t}\left(t\right)∥}_{1}^{2}dt\\ & +c\tau ∥B_{\epsilon }\left(t_{n}\right){∥}_{1}^{2}\int_{t_{n-1}}^{t_{n}}{∥B_{\epsilon t}\left(t\right)∥}_{1}^{2}dt,\\ 2|(R_{2}^{n},e_{B}^{n})|\leq & \frac{\nu }{6}∥A_{2}^{\frac{1}{2}}e_{B}^{n}{∥}^{2}+c\tau \int_{t_{n-1}}^{t_{n}}∥A_{2}^{-\frac{1}{2}}B_{\epsilon tt}{∥}^{2}dt+c\tau ∥u_{\epsilon }\left(t_{n}\right){∥}_{1}^{2}\int_{t_{n-1}}^{t_{n}}{∥B_{\epsilon t}\left(t\right)∥}_{1}^{2}dt. \end{array}$$Combining the above inequalities with (Equation 24), we obtain
(25)$$\begin{array}{cc} & ∥e_{u}^{n}{∥}^{2}+S∥e_{B}^{n}{∥}^{2}-(∥e_{u}^{n-1}{∥}^{2}+S∥e_{B}^{n-1}{∥}^{2})+\nu ∥A_{1\epsilon }^{\frac{1}{2}}e_{u}^{n}{∥}^{2}\tau +S\mu ∥A_{2}^{\frac{1}{2}}e_{B}^{n}{∥}^{2}\tau \\ & \;\leq cd_{n-1}(∥e_{u}^{n-1}{∥}^{2}+∥e_{B}^{n-1}{∥}^{2})\tau +c\tau^{2}\int_{t_{n-1}}^{t_{n}}(∥A_{1\epsilon }^{-\frac{1}{2}}u_{\epsilon tt}{∥}^{2}+∥A_{2}^{-\frac{1}{2}}B_{\epsilon tt}{∥}^{2})dt\\ & \;+c\tau^{2}(∥u_{\epsilon }\left(t_{n}\right){∥}_{1}^{2}+∥B_{\epsilon }\left(t_{n}\right){∥}_{1}^{2})\int_{t_{n-1}}^{t_{n}}(∥u_{\epsilon t}\left(t\right){∥}_{1}^{2}+∥B_{\epsilon t}\left(t\right){∥}_{1}^{2})dt, \end{array}$$
where $d_{n-1}=∥u_{\epsilon }\left(t_{n}\right){∥}_{2}^{2}+{∥B_{\epsilon }\left(t_{n}\right)∥}_{2}^{2}$. Summing (Equation 25) from n=1 to *m*, due to Theorem 2, we obtain
(26)$$\begin{array}{cc} & ∥e_{u}^{m}{∥}^{2}+S∥e_{B}^{m}{∥}^{2}+\tau \sum_{n=1}^{m}(\nu ∥A_{1\epsilon }^{\frac{1}{2}}e_{u}^{n}{∥}^{2}+S\mu ∥A_{2}^{\frac{1}{2}}e_{B}^{n}{∥}^{2})\leq c\tau \sum_{n=0}^{m-1}d_{n}(∥e_{u}^{n}{∥}^{2}+∥e_{B}^{n}{∥}^{2})+C\tau^{2}. \end{array}$$Then, by applying Lemma 2 to (Equation 26) and Theorem 2, we have
(27)$$\begin{array}{c} ∥e_{u}^{m}{∥}^{2}+S∥e_{B}^{m}{∥}^{2}+\tau \sum_{n=1}^{m}(\nu ∥A_{1\epsilon }^{\frac{1}{2}}e_{u}^{n}{∥}^{2}+S\mu ∥A_{2}^{\frac{1}{2}}e_{B}^{n}{∥}^{2})\leq C\tau^{2}\exp\left(\tau \sum_{n=0}^{m-1}d_{n}\right)\leq C\tau^{2}, \end{array}$$
for any 1≤m≤N. Using (Equation 27), (Equation 9) and Lemma 1, we obtain Theorem 5. □

**Theorem** **6.**
*Under Assumptions 1–3, we have*

$$\begin{array}{c} \underset{1\leq n\leq N}{\sup}\sigma^{\frac{1}{2}}\left(t_{n}\right)(∥u_{\epsilon }\left(t_{n}\right)-u_{\epsilon }^{n}{∥}_{1}+\sqrt{S\mu }∥B_{\epsilon }\left(t_{n}\right)-B_{\epsilon }^{n}{∥}_{1})+{\left(\tau \sum_{n=1}^{N}\sigma \left(t_{n}\right){∥p_{\epsilon }\left(t_{n}\right)-p_{\epsilon }^{n}∥}^{2}\right)}^{\frac{1}{2}}\leq C\tau . \end{array}$$



**Proof.** Taking the inner product of (Equation 20) with $2d_{t}e_{u}^{n}\tau$, (Equation 21) with $2d_{t}e_{B}^{n}\tau$, we deduce that
(28)$$\begin{array}{cc} & 2∥d_{t}e_{u}^{n}{∥}^{2}\tau +2∥d_{t}e_{B}^{n}{∥}^{2}\tau +\nu (∥A_{1\epsilon }^{\frac{1}{2}}e_{u}^{n}{∥}^{2}-∥A_{1\epsilon }^{\frac{1}{2}}e_{u}^{n-1}{∥}^{2}+∥A_{1\epsilon }^{\frac{1}{2}}\left(e_{u}^{n}-e_{u}^{n-1}\right){∥}^{2})\\ & \;+\mu (∥A_{2}^{\frac{1}{2}}e_{B}^{n}{∥}^{2}-∥A_{2}^{\frac{1}{2}}e_{B}^{n-1}{∥}^{2}+∥A_{2}^{\frac{1}{2}}\left(e_{B}^{n}-e_{B}^{n-1}\right){∥}^{2})+2b\left(e_{u}^{n-1},u_{\epsilon }\left(t_{n}\right),d_{t}e_{u}^{n}\right)\tau \\ & \;+2b\left(u_{\epsilon }\left(t_{n-1}\right),e_{u}^{n},d_{t}e_{u}^{n}\right)\tau +2b\left(e_{u}^{n-1},e_{u}^{n},d_{t}e_{u}^{n}\right)\tau +2S\left(e_{B}^{n-1}\times \nabla \times B_{\epsilon }\left(t_{n}\right),d_{t}e_{u}^{n}\right)\tau \\ & \;+2S\left(B_{\epsilon }\left(t_{n-1}\right)\times \nabla \times e_{B}^{n},d_{t}e_{u}^{n}\right)\tau +2S\left(e_{B}^{n-1}\times \nabla \times e_{B}^{n},d_{t}e_{u}^{n}\right)\tau +2\left(\nabla \times \left(u_{\epsilon }\left(t_{n}\right)\times e_{B}^{n-1}\right),d_{t}e_{B}^{n}\right)\tau \\ & \;+2\left(\nabla \times \left(e_{u}^{n}\times B_{\epsilon }\left(t_{n-1}\right)\right),d_{t}e_{B}^{n}\right)\tau +2\left(\nabla \times \left(e_{u}^{n}\times e_{B}^{n-1}\right),d_{t}e_{B}^{n}\right)\tau =2\left(R_{1}^{n},d_{t}e_{u}^{n}\right)\tau +2\left(R_{2}^{n},d_{t}e_{B}^{n}\right)\tau . \end{array}$$Using (Equation 7)–(Equation 12) and Lemma 1, we obtain
$$\begin{array}{cc} 2|b(e_{u}^{n-1},u_{\epsilon }\left(t_{n}\right),d_{t}e_{u}^{n})| & \leq \frac{1}{6}∥d_{t}e_{u}^{n}{∥}^{2}+c∥u_{\epsilon }\left(t_{n}\right){∥}_{2}^{2}∥A_{1\epsilon }^{\frac{1}{2}}e_{u}^{n-1}{∥}^{2},\\ 2|b(u_{\epsilon }\left(t_{n-1}\right),e_{u}^{n},d_{t}e_{u}^{n})| & \leq \frac{1}{6}∥d_{t}e_{u}^{n}{∥}^{2}+c∥u_{\epsilon }\left(t_{n-1}\right){∥}_{2}^{2}∥A_{1\epsilon }^{\frac{1}{2}}e_{u}^{n-1}{∥}^{2},\\ 2|b(e_{u}^{n-1},e_{u}^{n},d_{t}e_{u}^{n})| & \leq \frac{\nu }{6\tau }∥A_{1\epsilon }^{\frac{1}{2}}\left(e_{u}^{n}-e_{u}^{n-1}\right){∥}^{2}+c\frac{1}{\tau }∥e_{u}^{n}{∥}_{1}^{2}∥A_{1\epsilon }^{\frac{1}{2}}e_{u}^{n-1}{∥}^{2},\\ 2S|(e_{B}^{n-1}\times \nabla \times B_{\epsilon }\left(t_{n}\right),d_{t}e_{u}^{n})| & \leq \frac{1}{6}∥d_{t}e_{u}^{n}{∥}^{2}+c∥B_{\epsilon }\left(t_{n}\right){∥}_{2}^{2}∥A_{2}^{\frac{1}{2}}e_{B}^{n-1}{∥}^{2},\\ 2S|(B_{\epsilon }\left(t_{n-1}\right)\times \nabla \times e_{B}^{n},d_{t}e_{u}^{n})| & \leq \frac{1}{6}∥d_{t}e_{u}^{n}{∥}^{2}+c∥B_{\epsilon }\left(t_{n-1}\right){∥}_{2}^{2}∥A_{2}^{\frac{1}{2}}e_{B}^{n}{∥}^{2},\\ 2S|(e_{B}^{n-1}\times \nabla \times e_{B}^{n},d_{t}e_{u}^{n})| & \leq \frac{1}{6\tau }∥A_{1\epsilon }^{\frac{1}{2}}\left(e_{u}^{n}-e_{u}^{n-1}\right){∥}^{2}+c\frac{1}{\tau }∥e_{B}^{n}{∥}_{1}^{2}∥A_{1\epsilon }^{\frac{1}{2}}e_{B}^{n-1}{∥}^{2},\\ 2|(\nabla \times \left(u_{\epsilon }\left(t_{n}\right)\times e_{B}^{n-1}\right),d_{t}e_{B}^{n})| & \leq \frac{1}{6}∥d_{t}e_{B}^{n}{∥}^{2}+c∥u_{\epsilon }\left(t_{n}\right){∥}_{2}^{2}∥A_{2}^{\frac{1}{2}}e_{B}^{n-1}{∥}^{2},\\ 2|(\nabla \times \left(e_{u}^{n}\times B_{\epsilon }\left(t_{n-1}\right)\right),d_{t}e_{B}^{n})| & \leq \frac{1}{6}∥d_{t}e_{B}^{n}{∥}^{2}+c∥B_{\epsilon }\left(t_{n-1}\right){∥}_{2}^{2}∥A_{1\epsilon }^{\frac{1}{2}}e_{u}^{n}{∥}^{2},\\ 2|(\nabla \times \left(e_{u}^{n}\times e_{B}^{n-1}\right),d_{t}e_{B}^{n})| & \leq \frac{1}{6\tau }∥A_{2}^{\frac{1}{2}}\left(e_{B}^{n}-e_{B}^{n-1}\right){∥}^{2}+c\frac{1}{\tau }∥e_{u}^{n}{∥}_{1}^{2}∥A_{1\epsilon }^{\frac{1}{2}}e_{B}^{n-1}{∥}^{2}. \end{array}$$Similarly, we have
$$\begin{array}{cc} 2|(R_{1}^{n},d_{t}e_{u}^{n})|\leq & \frac{1}{6}∥d_{t}e_{u}^{n}{∥}^{2}+c\tau \int_{t_{n-1}}^{t_{n}}∥u_{\epsilon tt}{∥}^{2}dt+c\tau ∥u_{\epsilon }\left(t_{n}\right){∥}_{2}^{2}\int_{t_{n-1}}^{t_{n}}{∥u_{\epsilon t}\left(t\right)∥}_{1}^{2}dt\\ & +c\tau ∥B_{\epsilon }\left(t_{n}\right){∥}_{2}^{2}\int_{t_{n-1}}^{t_{n}}{∥B_{\epsilon t}\left(t\right)∥}_{1}^{2}dt,\\ 2|(R_{2}^{n},d_{t}e_{B}^{n})|\leq & \frac{1}{6}∥d_{t}e_{B}^{n}{∥}^{2}+c\tau \int_{t_{n-1}}^{t_{n}}∥B_{\epsilon tt}{∥}^{2}dt+c\tau ∥u_{\epsilon }\left(t_{n}\right){∥}_{2}^{2}\int_{t_{n-1}}^{t_{n}}{∥B_{\epsilon t}\left(t\right)∥}_{1}^{2}dt. \end{array}$$Combining the above inequalities with (Equation 28), and using Theorem 2, we can derive
$$\begin{array}{cc} & ∥d_{t}e_{u}^{n}{∥}^{2}\tau +∥d_{t}e_{B}^{n}{∥}^{2}\tau +\nu (∥A_{1\epsilon }^{\frac{1}{2}}e_{u}^{n}{∥}^{2}-∥A_{1\epsilon }^{\frac{1}{2}}e_{u}^{n-1}{∥}^{2})+\mu (∥A_{2}^{\frac{1}{2}}e_{B}^{n}{∥}^{2}-∥A_{2}^{\frac{1}{2}}e_{B}^{n-1}{∥}^{2})\\ & \;\leq c(∥A_{1\epsilon }^{\frac{1}{2}}e_{u}^{n-1}{∥}^{2}+∥A_{2}^{\frac{1}{2}}e_{B}^{n-1}{∥}^{2}+∥A_{1\epsilon }^{\frac{1}{2}}e_{u}^{n}{∥}^{2}+∥A_{2}^{\frac{1}{2}}e_{B}^{n}{∥}^{2})\tau +cd_{n-1}(∥A_{1\epsilon }^{\frac{1}{2}}e_{u}^{n-1}{∥}^{2}+∥A_{2}^{\frac{1}{2}}e_{B}^{n-1}{∥}^{2})\tau \\ & \;+c\tau^{2}\int_{t_{n-1}}^{t_{n}}(∥u_{\epsilon tt}{∥}^{2}+∥B_{\epsilon tt}{∥}^{2})dt+c\tau^{2}(∥u_{\epsilon }\left(t_{n}\right){∥}_{2}^{2}+∥B_{\epsilon }\left(t_{n}\right){∥}_{2}^{2})\int_{t_{n-1}}^{t_{n}}(∥u_{\epsilon t}\left(t\right){∥}_{1}^{2}+∥B_{\epsilon t}\left(t\right){∥}_{1}^{2})dt, \end{array}$$
where $d_{n-1}=\tau^{-1}(∥e_{u}^{n}{∥}_{1}^{2}+∥e_{B}^{n}{∥}_{1}^{2})$. Multiplying this inequality by $\sigma (t_{n})$ and taking the sum with respect to *n* from 1 to *m*, thanks to Theorems 2 and 5, we obtain
$$\begin{array}{cc} & \sigma \left(t_{m}\right)(\nu (∥A_{1\epsilon }^{\frac{1}{2}}e_{u}^{m}{∥}^{2}+\mu (∥A_{2}^{\frac{1}{2}}e_{B}^{m}{∥}^{2})+\tau \sum_{n=1}^{m}\sigma \left(t_{n}\right)(∥d_{t}e_{u}^{n}{∥}^{2}+∥d_{t}e_{B}^{n}{∥}^{2})\\ & \;\leq c\tau \sum_{n=0}^{m-1}d_{n}(∥A_{1\epsilon }^{\frac{1}{2}}e_{u}^{n}{∥}^{2}+∥A_{2}^{\frac{1}{2}}e_{B}^{n}{∥}^{2})+C\tau^{2}. \end{array}$$Then, by applying Lemma 2 to this inequality and using Theorem 5, we have
(29)$$\begin{array}{cc} & \sigma \left(t_{m}\right)(\nu (∥A_{1\epsilon }^{\frac{1}{2}}e_{u}^{m}{∥}^{2}+\mu (∥A_{2}^{\frac{1}{2}}e_{B}^{m}{∥}^{2})+\tau \sum_{n=1}^{m}\sigma \left(t_{n}\right)(∥d_{t}e_{u}^{n}{∥}^{2}+∥d_{t}e_{B}^{n}{∥}^{2})\leq C\tau^{2}, \end{array}$$
for all 1≤m≤N.Finally, using (Equation 7)–(Equation 12) and the LBB condition (cf. [4,13])
(30)$$\begin{array}{c} \beta ∥q∥\leq \underset{v\in X}{\sup}\frac{\left(\nabla \cdot v,q\right)}{∥\nabla v∥},\;\forall q\in M, \end{array}$$
we derive
$$\begin{array}{cc} \beta ∥p_{\epsilon }\left(t_{n}\right)-p_{\epsilon }^{n}∥ & \leq c∥d_{t}e_{u}^{n}∥+\nu ∥e_{u}^{n}{∥}_{1}+c∥e_{u}^{n-1}{∥}_{1}∥u_{\epsilon }\left(t_{n}\right){∥}_{1}+c∥u_{\epsilon }^{n-1}{∥}_{1}{∥e_{u}^{n}∥}_{1}\\ & +c∥e_{B}^{n-1}{∥}_{1}∥\nabla \times B_{\epsilon }\left(t_{n}\right)∥+c∥B_{\epsilon }^{n-1}{∥}_{1}∥\nabla \times e_{B}^{n}∥+c∥R_{1}^{n}∥\\ & \leq c∥d_{t}e_{u}^{n}∥+\nu ∥e_{u}^{n}{∥}_{1}+c∥e_{u}^{n-1}{∥}_{1}∥u_{\epsilon }\left(t_{n}\right){∥}_{1}+c∥u_{\epsilon }^{n-1}{∥}_{1}{∥e_{u}^{n}∥}_{1}\\ & +c∥e_{B}^{n-1}{∥}_{1}∥B_{\epsilon }\left(t_{n}\right){∥}_{1}+c∥B_{\epsilon }^{n-1}{∥}_{1}∥e_{B}^{n}{∥}_{1}+c\tau \int_{t_{n-1}}^{t_{n}}{∥u_{\epsilon tt}∥}^{2}dt\\ & +c\tau ∥u_{\epsilon }\left(t_{n}\right){∥}_{2}^{2}\int_{t_{n-1}}^{t_{n}}∥u_{\epsilon t}\left(t\right){∥}_{1}^{2}dt+c\tau ∥B_{\epsilon }\left(t_{n}\right){∥}_{2}^{2}\int_{t_{n-1}}^{t_{n}}{∥B_{\epsilon t}\left(t\right)∥}_{1}^{2}dt. \end{array}$$Multiplying this inequality by $\sigma (t_{n})$ and taking the sum with respect to *n* from 1 to *m*, due to Theorems 2 and 5 and (Equation 29), we obtain
(31)$$\begin{array}{cc} \tau \sum_{n=1}^{m}\sigma \left(t_{n}\right){∥p_{\epsilon }\left(t_{n}\right)-p_{\epsilon }^{n}∥}^{2}\; & \leq C\tau^{2}, \end{array}$$
for all 1≤m≤N. Using (Equation 29), (Equation 31), (Equation 9), and Lemma 1, we obtain Theorem 6. □

## 4. PFEM for the MHD Equations

We further consider a spatial discretization for the penalty MHD system of time discretization in this section (cf. [4]). $J_{h}$ is a family of quasi-uniformly regular partitions of D into triangles or tetrahedron elements *K* with the diameter $h_{K}$. Let the mesh size $h=\max_{K\in J_{h}}h_{K}$. We give three finite element spaces $X_{h}$, $M_{h}$, $W_{h}$ with $X_{h}\subset X$, $M_{h}\subset M$ and $W_{h}\subset W$.

Let $\rho_{h}:M\to M_{h}$ denote the $L^{2}$-orthogonal projector which is defined by
(32)$$\begin{array}{c} \left(\rho_{h}q,q_{h}\right)=\left(q,q_{h}\right),\;\forall q\in M,q_{h}\in M_{h}. \end{array}$$

**Assumption** **4.**
*The finite element space $\left(X_{h},M_{h}\right)$ satisfies the discrete LBB condition (cf. [4,22])*

(33)
$$\begin{array}{c} \underset{v_{h}\in X_{h}}{\sup}\frac{d(v_{h},q_{h})}{∥\nabla v_{h}∥}\geq \beta_{1}∥q_{h}∥,\;q_{h}\in M_{h}, \end{array}$$

*where $\beta_{1}$ is a positive constant depending on D. For each $v\in H^{2}\left(D\right)\cap X$, $q\in H^{1}\left(D\right)\cap M,C\in H^{2}\left(D\right)\cap W$, there exist $\pi_{h}v\in X_{h}$, $\rho_{h}q\in M_{h}$ and $J_{h}C\in W_{h}$ such that*

(34)
$$\begin{array}{cc} & \left(\nabla \cdot \left(v-\pi_{h}v\right),q_{h}\right)=0,\;q_{h}\in M_{h}, \end{array}$$


(35)
$$\begin{array}{cc} & ∥\nabla \left(v-\pi_{h}v\right){∥\leq ch∥v∥}_{2},\;∥q-\rho_{h}{q∥\leq ch∥q∥}_{1},\;∥\nabla \left(C-J_{h}C\right){∥\leq ch∥C∥}_{2}, \end{array}$$

*together with the inverse inequalities*

(36)
$$\begin{array}{c} ∥\nabla v_{h}∥\leq ch^{-1}∥v_{h}∥,\;v_{h}\in X_{h},\;∥\nabla C_{h}∥\leq ch^{-1}∥C_{h}∥,\;C_{h}\in W_{h}. \end{array}$$



Next, to obtain an approximation of (u,p,B), we consider the following finite element pairs:$$\begin{array}{c} X_{h}={\left(P_{1,h}^{b}\right)}^{d}\cap X,\\ M_{h}=\left\{q_{h}\in C^{0}\left(D\right)\cap M:q_{h}{|}_{K}\in P_{1}\left(K\right),\forall K\in J_{h}\right\},\\ W_{h}=\left\{C_{h}\in C^{0}\left(D\right)\cap W:C_{h}{|}_{K}\in P_{1}{\left(K\right)}^{d},\forall K\in J_{h}\right\}, \end{array}$$
where
$$\begin{array}{c} P_{1,h}^{b}=\left\{v_{h}\in C^{0}\left(D\right):v_{h}{|}_{K}\in P_{1}\left(K\right)⊕span\left\{b\right\},\forall K\in J_{h}\right\}, \end{array}$${b} is a bubble function. Let $b\in H_{0}^{1}\left(K\right)$ take the value 1 at the barycentre of *K* and satisfy 0≤b(x)≤1, which is called a “bubble function” (cf. [22]). Furthermore, we denote the discrete subspace $X_{0h}$ of $X_{0}$ as
$$\begin{array}{c} X_{0h}=\left\{v_{h}\in X_{h}:d\left(v_{h},q_{h}\right)=0,q_{h}\in M_{h}\right\}. \end{array}$$

The finite element approximation for (Equation 16) based on $X_{h}\times M_{h}\times W_{h}$ is given as follows: find $\left(u_{\epsilon h}^{n},p_{\epsilon h}^{n},B_{\epsilon h}^{n}\right)$ such that for all 1≤n≤N and $\left(v_{h},q_{h},C_{h}\right)\in X_{h}\times M_{h}\times W_{h}$,
(37)$$\begin{array}{cc} & \left(d_{t}u_{\epsilon h}^{n},v\right)+\nu \left(\nabla u_{\epsilon h}^{n},\nabla v\right)+b\left(u_{\epsilon h}^{n-1},u_{\epsilon h}^{n},v\right)-\left(\nabla \cdot v,p_{\epsilon h}^{n}\right)+\left(\nabla \cdot u_{\epsilon h}^{n},q\right)+S\left(B_{\epsilon h}^{n-1}\times \nabla \times B_{\epsilon h}^{n},v\right)\\ & \;+\frac{\epsilon }{\nu }\left(p_{\epsilon h}^{n},q\right)=\left(f\left(t_{n}\right),v\right), \end{array}$$
(38)$$\begin{array}{cc} & \left(d_{t}B_{\epsilon h}^{n},C\right)+\mu \left(\nabla \times B_{\epsilon h}^{n},\nabla \times C\right)+\mu \left(\nabla \cdot B_{\epsilon h}^{n},\nabla \cdot C\right)-\left(u_{\epsilon h}^{n}\times B_{\epsilon h}^{n-1},\nabla \times C\right)=\left(g\left(t_{n}\right),C\right), \end{array}$$
(39)$$\begin{array}{cc} & u_{\epsilon h}^{0}=r_{1h}u^{0},\;B_{\epsilon h}^{0}=r_{2h}B^{0}, \end{array}$$
where $r_{1h}:L^{2}\left(D\right)\to X_{0h}$ and $r_{2h}:L^{2}\left(D\right)\to W_{h}$ are $L^{2}$-orthogonal projectors. According to (Equation 34)–(Equation 36), these operators satisfy the following properties (cf. [3,4,22]): (40)$$\begin{array}{cc} & ∥v-r_{1h}v∥+h∥\nabla \left(v-r_{1h}v\right)∥\leq ch^{2}∥A_{1}v∥,\;\forall v\in H^{2}\left(D\right)\cap X, \end{array}$$(41)$$\begin{array}{cc} & ∥w-r_{2h}w∥+h∥\nabla \left(w-r_{2h}w\right)∥\leq ch^{i}{∥w∥}_{i},\;\forall w\in H^{2}\left(D\right)\cap W_{0},\;i=1,2. \end{array}$$

Here, we define the discrete Stokes operator $A_{1h}=-r_{1h}\Delta_{h}$, which is defined by (see [4])
$$\begin{array}{c} \left(-\Delta_{h}u_{h},v_{h}\right)=\left(\nabla u_{h},\nabla v_{h}\right),\;u_{h},v_{h}\in X_{h}, \end{array}$$
its discrete norm $∥v_{h}{∥}_{k,2}=∥A_{1h}^{\frac{k}{2}}v_{h}∥$ of the k∈R order can be defined, where
$$\begin{array}{c} ∥v_{h}{∥}_{1,2}=∥\nabla v_{h}∥,\;∥v_{h}{∥}_{2,2}=∥A_{1h}v_{h}∥,\;∥v_{h}{∥}_{-1,2}=∥A_{1h}^{-\frac{1}{2}}v_{h}∥=\underset{w_{h}\in X_{h}}{\sup}\frac{\left(v_{h},w_{h}\right)}{∥A_{1h}^{\frac{1}{2}}w_{h}∥},\;\forall v_{h}\in X_{0h}. \end{array}$$Meanwhile, we define the discrete operator $A_{2h}B_{h}=r_{2h}\left(\nabla_{h}\times \nabla \times B_{h}+\nabla_{h}\nabla \cdot B_{h}\right)\in W_{h}$ as follows:$$\begin{array}{c} \left(A_{2h}B_{h},C_{h}\right)=\left(A_{2h}^{\frac{1}{2}}B_{h},A_{2h}^{\frac{1}{2}}C_{h}\right)=\left(\nabla \times B_{h},\nabla \times C_{h}\right)+\left(\nabla \cdot B_{h},\nabla \cdot C_{h}\right), \end{array}$$
and its discrete norm $∥B_{h}{∥}_{k}=∥A_{2h}^{\frac{k}{2}}B_{h}∥$ of the k∈R order can be defined, where
$$\begin{array}{c} ∥B_{h}{∥}_{1,2}=∥A_{2h}^{\frac{1}{2}}B_{h}∥,\;∥B_{h}{∥}_{2,2}=∥A_{2h}B_{h}∥,\;∥B_{h}{∥}_{-1,2}=∥A_{2h}^{-\frac{1}{2}}B_{h}∥=\underset{C_{h}\in W_{h}}{\sup}\frac{\left(B_{h},C_{h}\right)}{∥A_{2h}^{\frac{1}{2}}C_{h}∥},\;\forall B_{h}\in W_{h}. \end{array}$$

To obtain the error analysis of the scheme in the following section, we need the following discrete estimates which are obtained from [4,7]).

**Lemma** **3.**
*The estimates of $v_{h}$ and $C_{h}$ are as follows:*

$$\begin{array}{c} ∥v_{h}{∥}_{L^{6}}\leq ∥v_{h}{∥}_{1,2},\;∥v_{h}{∥}_{L^{3}}\leq c∥v_{h}{∥}^{\frac{1}{2}}{∥v_{h}∥}_{1,2}^{\frac{1}{2}},\\ ∥\nabla v_{h}{∥}_{L^{6}}\leq ∥v_{h}{∥}_{2,2},\;∥\nabla v_{h}{∥}_{L^{3}}+∥v_{h}{∥}_{L^{\infty }}\leq c∥v_{h}{∥}_{1,2}^{\frac{1}{2}}{∥v_{h}∥}_{2,2}^{\frac{1}{2}},\;\forall v_{h}\in X_{0h},\\ ∥C_{h}{∥}_{L^{6}}\leq ∥C_{h}{∥}_{1,2},\;∥C_{h}{∥}_{L^{3}}\leq c∥C_{h}{∥}^{\frac{1}{2}}{∥C_{h}∥}_{1,2}^{\frac{1}{2}},\\ ∥\nabla C_{h}{∥}_{L^{6}}\leq ∥C_{h}{∥}_{2,2},\;∥\nabla C_{h}{∥}_{L^{3}}+∥C_{h}{∥}_{L^{\infty }}\leq c∥C_{h}{∥}_{1,2}^{\frac{1}{2}}{∥C_{h}∥}_{2,2}^{\frac{1}{2}},\;\forall C_{h}\in W_{h}. \end{array}$$



For the finite element space $X_{h}\times M_{h}\times W_{h}$ given above, problems (Equation 37) and (Equation 38) allow us to calculate velocity and pressure separately, i.e., (Equation 37) and (Equation 38) can be reduced as follows: find $\left(u_{\epsilon h}^{n},p_{\epsilon h}^{n},B_{\epsilon h}^{n}\right)\in X_{h}\times M_{h}\times W_{h}$ such that
(42)$$\left\{\begin{array}{c} \left(d_{t}u_{\epsilon h}^{n},v\right)+\nu \left(A_{1h}^{\frac{1}{2}}u_{\epsilon h}^{n},A_{1h}^{\frac{1}{2}}v\right)+b\left(u_{\epsilon h}^{n-1},u_{\epsilon h}^{n},v\right)+S\left(B_{\epsilon h}^{n-1}\times \nabla \times B_{\epsilon h}^{n},v\right)\\ \;+\frac{\nu }{\epsilon }\left(\nabla \cdot v,\rho_{h}\nabla \cdot u_{\epsilon h}^{n}\right)=\left(f\left(t_{n}\right),v\right),\\ \left(d_{t}B_{\epsilon h}^{n},C\right)+\mu \left(A_{2h}^{\frac{1}{2}}B_{\epsilon h}^{n},A_{2h}^{\frac{1}{2}}C\right)-\left(u_{\epsilon h}^{n}\times B_{\epsilon h}^{n-1},\nabla \times C\right)=\left(g\left(t_{n}\right),C\right),\\ p_{\epsilon h}^{n}=-\frac{\nu }{\epsilon }\rho_{h}\nabla \cdot u_{\epsilon h}^{n}. \end{array}\right.$$In the following, we give the algebraic matrix form for the 2D case, and the algebraic matrix form for the 3D case is similar. In order to analyze the detailed form of the coefficient matrix for this scheme, we write the fluid velocity and magnetic field vectors
$$u_{\epsilon h}^{n}=\left(u_{1\epsilon h}^{n},u_{2\epsilon h}^{n}\right),\;B_{\epsilon h}^{n}=\left(B_{1\epsilon h}^{n},B_{2\epsilon h}^{n}\right),$$
and the corresponding test function vectors
$$\begin{array}{c} v_{h}^{n}=\left(v_{1h}^{n},v_{2h}^{n}\right),\;C_{h}^{n}=\left(C_{1h}^{n},C_{2h}^{n}\right), \end{array}$$
then, we expand (Equation 42) and obtain that

Step 1. Find $\left(u_{\epsilon h}^{n},B_{\epsilon h}^{n}\right)$ from
$$\begin{array}{c} \left\{\begin{array}{c} \tau^{-1}\left(u_{1\epsilon h}^{n},v_{1h}\right)+\nu \left(\partial_{x}u_{1\epsilon h}^{n},\partial_{x}v_{1h}\right)+\nu \left(\partial_{y}u_{1\epsilon h}^{n},\partial_{y}v_{1h}\right)+\left(u_{1\epsilon h}^{n-1}\partial_{x}u_{1\epsilon h}^{n},v_{1h}\right)\\ \;+\left(u_{2\epsilon h}^{n-1}\partial_{y}u_{1\epsilon h}^{n},v_{1h}\right)+S\left(B_{2\epsilon h}^{n-1}\partial_{x}B_{2\epsilon h}^{n},v_{1h}\right)-S\left(B_{2\epsilon h}^{n-1}\partial_{y}B_{1\epsilon h}^{n},v_{1h}\right)\\ \;+\frac{\nu }{\epsilon }\left(\rho_{h}\left(\partial_{x}u_{1\epsilon h}^{n}+\partial_{y}u_{2\epsilon h}^{n}\right),\partial_{x}v_{1h}\right)=\left(f_{1}\left(t_{n}\right),v_{1h}\right)+\tau^{-1}\left(u_{1\epsilon h}^{n-1},v_{1h}\right),\\ \tau^{-1}\left(u_{2\epsilon h}^{n},v_{2h}\right)+\nu \left(\partial_{x}u_{2\epsilon h}^{n},\partial_{x}v_{2h}\right)+\nu \left(\partial_{y}u_{2\epsilon h}^{n},\partial_{y}v_{2h}\right)+\left(u_{1\epsilon h}^{n-1}\partial_{x}u_{2\epsilon h}^{n},v_{2h}\right)\\ \;+\left(u_{2\epsilon h}^{n-1}\partial_{y}u_{2\epsilon h}^{n},v_{2h}\right)-S\left(B_{1\epsilon h}^{n-1}\partial_{x}B_{2\epsilon h}^{n},v_{2h}\right)+S\left(B_{1\epsilon h}^{n-1}\partial_{y}B_{1\epsilon h}^{n},v_{2h}\right)\\ \;+\frac{\nu }{\epsilon }\left(\rho_{h}\left(\partial_{y}u_{2\epsilon h}^{n}+\partial_{y}u_{2\epsilon h}^{n}\right),\partial_{x}v_{1h}\right)=\left(f_{2}\left(t_{n}\right),v_{2h}\right)+\tau^{-1}\left(u_{2\epsilon h}^{n-1},v_{2h}\right),\\ \tau^{-1}\left(B_{1\epsilon h}^{n},C_{1h}\right)-\eta \left(\partial_{x}B_{2\epsilon h}^{n}-\partial_{y}B_{1\epsilon h}^{n},\partial_{y}C_{1h}\right)+\eta \left(\partial_{x}B_{1\epsilon h}^{n}+\partial_{y}B_{2\epsilon h}^{n},\partial_{x}C_{1h}\right)\\ \;+\left(u_{1\epsilon h}^{n}B_{2\epsilon h}^{n-1}-u_{2\epsilon h}^{n}B_{1\epsilon h}^{n-1},\partial_{y}C_{1h}\right)=\left(g_{1}\left(t_{n}\right),C_{1h}\right)+\tau^{-1}\left(B_{1\epsilon h}^{n-1},C_{1h}\right),\\ \tau^{-1}\left(B_{2\epsilon h}^{n},C_{2h}\right)+\eta \left(\partial_{x}B_{2\epsilon h}^{n}-\partial_{y}B_{1\epsilon h}^{n},\partial_{x}C_{2h}\right)+\eta \left(\partial_{x}B_{1\epsilon h}^{n}+\partial_{y}B_{2\epsilon h}^{n},\partial_{y}C_{2h}\right)\\ \;-\left(u_{1\epsilon h}^{n}B_{2\epsilon h}^{n-1}-u_{2\epsilon h}^{n}B_{1\epsilon h}^{n-1},\partial_{x}C_{2h}\right)=\left(g_{2}\left(t_{n}\right),C_{2h}\right)+\tau^{-1}\left(B_{2\epsilon h}^{n-1},C_{2h}\right). \end{array}\right. \end{array}$$

Step 2. Find $p_{\epsilon h}^{n}$ from
$$\begin{array}{c} p_{\epsilon h}^{n}=-\frac{\nu }{\epsilon }\rho_{h}\left(\partial_{x}u_{1\epsilon h}^{n}+\partial_{y}u_{2\epsilon h}^{n}\right). \end{array}$$Here, $\partial_{x}v_{1h}=\partial v_{1h}/\partial x$ and $\partial_{y}v_{2h}=\partial v_{2h}/\partial y$. Next, we assume the spaces $X_{h}$ and $W_{h}$ are combined with the basis functions
$$\begin{array}{c} X_{h}=span\left\{\varphi_{i}:i=1,\cdots ,N\right\},\;W_{h}=span\left\{\psi_{i}:i=1,\cdots ,M\right\}, \end{array}$$
where N and M denote the number of the basis functions in each of spaces. Then,
$$\begin{array}{cc} & u_{l\epsilon h}^{n}=\sum_{i=1}^{N}u_{l\epsilon h,i}^{n}\varphi_{i},\;v_{lh}=\varphi_{j},\;l=1,2,\;j=1,\cdots ,N,\\ & B_{l\epsilon h}^{n}=\sum_{i=1}^{M}B_{l\epsilon h,i}^{n}\psi_{i},\;C_{lh}=\psi_{j},\;l=1,2,\;j=1,\cdots ,M. \end{array}$$Next, we show the relationship between the $\left(u_{\epsilon h}^{n-1},B_{\epsilon h}^{n-1}\right)$ and $\left(u_{\epsilon h}^{n},B_{\epsilon h}^{n}\right)$. Apparently, step 1 of the fully discrete PFEM generates an algebraic system as follows:$$\left[\begin{array}{cc} A & B_{1}\\ B_{2} & C \end{array}\right]\left[\begin{array}{c} u\\ B \end{array}\right]=\left[\begin{array}{c} F\\ G \end{array}\right],$$
where
$$\begin{array}{c} u={\left(u_{1\epsilon h,i=1,N}^{n},u_{2\epsilon h,i=1,N}^{n}\right)}^{T},\;B={\left(B_{1\epsilon h,i=1,M}^{n},B_{2\epsilon h,i=1,M}^{n}\right)}^{T},\\ F={\left(\left(f_{1}\left(t_{n}\right),\varphi_{j}\right)+\tau^{-1}\left(u_{1\epsilon h,i}^{n-1},\varphi_{j}\right),\left(f_{2}\left(t_{n}\right),\varphi_{j}\right)+\tau^{-1}\left(u_{2\epsilon h,i}^{n-1},\varphi_{j}\right)\right)}^{T},\\ G={\left(\left(g_{1}\left(t_{n}\right),\psi_{j}\right)+\tau^{-1}\left(B_{1\epsilon h,i}^{n-1},\psi_{j}\right),\left(g_{2}\left(t_{n}\right),\psi_{j}\right)+\tau^{-1}\left(B_{2\epsilon h,i}^{n-1},\psi_{j}\right)\right)}^{T}. \end{array}$$Specifically, detailed calculation for PFEM gives
$$A=\left[\begin{array}{cc} a_{ij}+b_{ij} & c_{ij}\\ d_{ij} & a_{ij}+e_{ij} \end{array}\right],B_{1}=\left[\begin{array}{cc} g_{ij} & f_{ij}\\ k_{ij} & h_{ij} \end{array}\right],C=\left[\begin{array}{cc} l_{ij} & m_{ij}\\ m_{ij}^{T} & l_{ij} \end{array}\right],B_{2}=-S^{-1}B_{1}^{T},$$
where
$$\begin{array}{cc} & a_{ij}=\tau^{-1}\left(\varphi_{i},\varphi_{j}\right)+\nu \left(\left(\partial_{x}\varphi_{i},\partial_{x}\varphi_{j}\right)+\left(\partial_{y}\varphi_{i},\partial_{y}\varphi_{j}\right)\right)+\left(u_{1\epsilon h}^{n-1}\partial_{x}\varphi_{i}+u_{2\epsilon h}^{n-1}\partial_{y}\varphi_{i},\varphi_{j}\right),\\ & b_{ij}=\frac{\nu }{\epsilon }\left(\rho_{h}\partial_{x}\varphi_{i},\partial_{x}\varphi_{j}\right),\;c_{ij}=\frac{\nu }{\epsilon }\left(\rho_{h}\partial_{y}\varphi_{i},\partial_{x}\varphi_{j}\right),\;d_{ij}=\frac{\nu }{\epsilon }\left(\rho_{h}\partial_{x}\varphi_{i},\partial_{y}\varphi_{j}\right),\\ & e_{ij}=\frac{\nu }{\epsilon }\left(\rho_{h}\partial_{y}\varphi_{i},\partial_{y}\varphi_{j}\right),\;f_{ij}=S\left(B_{2\epsilon h}^{n-1}\partial_{x}\psi_{i},\varphi_{j}\right),\;g_{ij}=-S\left(B_{2\epsilon h}^{n-1}\partial_{y}\psi_{i},\varphi_{j}\right),\\ & l_{ij}=\tau^{-1}\left(\psi_{i},\psi_{j}\right)+\eta \left(\partial_{y}\psi_{i},\partial_{y}\psi_{j}\right)+\eta \left(\partial_{x}\psi_{i},\partial_{x}\psi_{j}\right),\;h_{ij}=-S\left(B_{1\epsilon h}^{n-1}\partial_{x}\psi_{i},\varphi_{j}\right),\\ & k_{ij}=S\left(B_{1\epsilon h}^{n-1}\partial_{y}\psi_{i},\varphi_{j}\right),\;m_{ij}=\eta \left(\partial_{y}\psi_{i},\partial_{x}\psi_{j}\right)-\eta \left(\partial_{x}\psi_{i},\partial_{y}\psi_{j}\right). \end{array}$$Then, we obtain $p_{\epsilon }^{n}$ by step 2.

Arguing in exactly the same way as in the proof of Theorem 4, using (Equation 9) and Lemma 1, we obtain a priori bound of schemes (Equation 37)–(Equation 39).

**Theorem** **7.**
*Under Assumptions 1–3, we have*

$$\begin{array}{c} ∥u_{\epsilon h}^{m}{∥}^{2}+S∥B_{\epsilon h}^{m}{∥}^{2}+\tau \sum_{n=1}^{m}(\nu ∥A_{1h}^{\frac{1}{2}}u_{\epsilon h}^{n}{∥}^{2}+S\mu ∥A_{2h}^{\frac{1}{2}}B_{\epsilon h}^{n}{∥}^{2})+\tau \sum_{n=1}^{m}(∥d_{t}u_{\epsilon h}^{n}{∥}^{2}+S∥d_{t}B_{\epsilon h}^{n}{∥}^{2})\leq C, \end{array}$$

*for all 1≤m≤N.*


## 5. Error Analysis for the Fully Discrete Euler Semi-Implicit Scheme

In this section, we establish the error estimates for $\left(u_{\epsilon h}^{n},p_{\epsilon h}^{n},B_{\epsilon h}^{n}\right)$ of the fully discrete Euler semi-implicit scheme (Equation 37)–(Equation 39). To this end, subtracting (Equation 37)–(Equation 38) from (Equation 16), we have
(43)$$\begin{array}{cc} & \left(d_{t}\left(u_{\epsilon }^{n}-u_{\epsilon h}^{n}\right),v\right)+\nu \left(\nabla \left(u_{\epsilon }^{n}-u_{\epsilon h}^{n}\right),\nabla v\right)+b\left(u_{\epsilon }^{n-1}-u_{\epsilon h}^{n-1},u_{\epsilon h}^{n},v\right)+b\left(u_{\epsilon h}^{n-1},u_{\epsilon }^{n}-u_{\epsilon h}^{n},v\right)\\ & \;-\left(\nabla \cdot v,p_{\epsilon }^{n}-p_{\epsilon h}^{n}\right)+\left(\nabla \cdot \left(u_{\epsilon }^{n}-u_{\epsilon h}^{n}\right),q\right)+S\left(\left(B_{\epsilon }^{n-1}-B_{\epsilon h}^{n-1}\right)\times \nabla \times B_{\epsilon h}^{n},v\right)\\ & \;+S\left(B_{\epsilon h}^{n-1}\times \nabla \times \left(B_{\epsilon }^{n}-B_{\epsilon h}^{n}\right),v\right)+\frac{\epsilon }{\nu }\left(p_{\epsilon }^{n}-p_{\epsilon h}^{n},q\right)=\left(f\left(t_{n}\right),v\right), \end{array}$$
(44)$$\begin{array}{cc} & \left(d_{t}\left(B_{\epsilon }^{n}-B_{\epsilon h}^{n}\right),C\right)+\mu \left(\nabla \times \left(B_{\epsilon }^{n}-B_{\epsilon h}^{n}\right),\nabla \times C\right)+\mu \left(\nabla \cdot \left(B_{\epsilon }^{n}-B_{\epsilon h}^{n}\right),\nabla \cdot C\right)\\ & \;-\left(\left(u_{\epsilon }^{n}-u_{\epsilon h}^{n}\right)\times B_{\epsilon h}^{n-1},\nabla \times C\right)-\left(u_{\epsilon h}^{n}\times \left(B_{\epsilon }^{n-1}-B_{\epsilon h}^{n-1}\right),\nabla \times C\right)=\left(g\left(t_{n}\right),C\right). \end{array}$$

In order to derive estimates of the error, we need the following regularity results.

**Lemma** **4.**
*Under Assumptions 1–3, we have the following estimates:*

$$\begin{array}{cc} & ∥A_{1\epsilon }^{\frac{1}{2}}u_{\epsilon }^{m}{∥}^{2}+∥A_{2}^{\frac{1}{2}}B_{\epsilon }^{m}{∥}^{2}+\tau \sum_{n=1}^{m}(∥A_{1\epsilon }u_{\epsilon }^{n}{∥}^{2}+∥A_{2}B_{\epsilon }^{n}{∥}^{2}+∥p_{\epsilon }^{n}{∥}_{1}^{2})\leq C,\\ & ∥A_{1\epsilon }u_{\epsilon }^{m}{∥}^{2}+∥A_{2}B_{\epsilon }^{m}{∥}^{2}+∥p_{\epsilon }^{m}{∥}_{1}^{2}+∥d_{t}u_{\epsilon }^{m}{∥}^{2}+∥d_{t}B_{\epsilon }^{m}{∥}^{2}+\tau \sum_{n=1}^{m}(∥A_{1\epsilon }^{\frac{1}{2}}d_{t}u_{\epsilon }^{n}{∥}^{2}+∥A_{2}^{\frac{1}{2}}d_{t}B_{\epsilon }^{n}{∥}^{2})\leq C,\\ & \sigma \left(t_{m}\right)(∥A_{1\epsilon }^{\frac{1}{2}}d_{t}u_{\epsilon }^{m}{∥}^{2}+∥A_{2}^{\frac{1}{2}}d_{t}B_{\epsilon }^{m}{∥}^{2})+\tau \sum_{n=1}^{m}\sigma \left(t_{n}\right)(∥A_{1\epsilon }d_{t}u_{\epsilon }^{n}{∥}^{2}+∥A_{2}d_{t}B_{\epsilon }^{n}{∥}^{2}+∥d_{t}p_{\epsilon }^{n}{∥}_{1}^{2})\leq C, \end{array}$$

*for all 1≤n≤N.*


We refer to [4,32] for the proof of these results.

**Theorem** **8.**
*Under Assumptions 1–3, we have*

$$\begin{array}{c} ∥u_{\epsilon }^{m}-u_{\epsilon h}^{m}∥+\sqrt{S}∥B_{\epsilon }^{m}-B_{\epsilon h}^{m}∥+\tau^{\frac{1}{2}}\sum_{n=1}^{m}(\sqrt{\nu }∥\nabla \left(u_{\epsilon }^{n}-u_{\epsilon h}^{n}\right)∥+\sqrt{S\mu }∥\nabla \left(B_{\epsilon }^{n}-B_{\epsilon h}^{n}\right)∥)\leq Ch. \end{array}$$



**Proof.** Setting $\eta_{u}^{n}=r_{1h}u_{\epsilon }^{n}-u_{\epsilon h}^{n}$, $\eta_{p}^{n}=\rho p_{\epsilon }^{n}-p_{\epsilon h}^{n}$ and $\eta_{B}^{n}=r_{2h}B_{\epsilon }^{n}-B_{\epsilon h}^{n}$ and taking $\left(v_{h},q_{h}\right)=2\left(\eta_{u}^{n},\eta_{p}^{n}\right)\tau$ in (Equation 43), $C_{h}=2S\eta_{B}^{n}\tau$ in (Equation 44), thanks to (Equation 4) and (Equation 10), we deduce that
(45)$$\begin{array}{cc} & ∥\eta_{u}^{n}{∥}^{2}-∥\eta_{u}^{n-1}{∥}^{2}+S(∥\eta_{B}^{n}{∥}^{2}-∥\eta_{B}^{n-1}{∥}^{2})+∥d_{t}\eta_{u}^{n}{∥}^{2}\tau^{2}+S{∥d_{t}\eta_{B}^{n}∥}^{2}\tau^{2}\\ & \;+\nu ∥A_{1h}^{\frac{1}{2}}\eta_{u}^{n}{∥}^{2}\tau +\nu ∥A_{1h}^{\frac{1}{2}}\left(u_{\epsilon }^{n}-u_{\epsilon h}^{n}\right){∥}^{2}\tau +S\mu ∥A_{2h}^{\frac{1}{2}}\eta_{B}^{n}{∥}^{2}+S\mu ∥A_{2h}^{\frac{1}{2}}\left(B_{\epsilon }^{n}-B_{\epsilon h}^{n}\right){∥}^{2}\tau \\ & \;+\frac{\epsilon }{\nu }(∥\eta_{p}^{n}{∥}^{2}+∥p_{\epsilon }-p_{\epsilon }^{n}{∥}^{2})\tau +2b\left(u_{\epsilon }^{n-1}-u_{\epsilon h}^{n-1},u_{\epsilon }^{n},\eta_{u}^{n}\right)\tau +2b\left(u_{\epsilon h}^{n-1},u_{\epsilon }^{n}-r_{1h}u_{\epsilon }^{n},\eta_{u}^{n}\right)\tau \\ & \;+2S\left(\left(B_{\epsilon }^{n-1}-B_{\epsilon h}^{n-1}\right)\times \nabla \times B_{\epsilon }^{n},\eta_{u}^{n}\right)\tau -2S\left(u_{\epsilon }^{n}\times \left(B_{\epsilon }^{n-1}-B_{\epsilon h}^{n-1}\right),\nabla \times \eta_{B}^{n}\right)\tau \\ & \;+2S\left(B_{\epsilon h}^{n-1}\times \nabla \times \left(B_{\epsilon }^{n}-r_{2h}B_{\epsilon }^{n}\right),\eta_{u}^{n}\right)\tau -2S\left(\left(u_{\epsilon }^{n}-r_{1h}u_{\epsilon }^{n}\right)\times B_{\epsilon }^{n-1},\nabla \times \eta_{B}^{n}\right)\tau \\ & \;+2S\left(\left(u_{\epsilon }^{n}-r_{1h}u_{\epsilon }^{n}\right)\times \left(B_{\epsilon }^{n-1}-B_{\epsilon h}^{n-1}\right),\nabla \times \eta_{B}^{n}\right)\tau =\nu ∥A_{1}^{\frac{1}{2}}\left(u_{\epsilon }^{n}-r_{1h}u_{\epsilon }^{n}\right){∥}^{2}\tau \\ & \;+S\mu ∥A_{2}^{\frac{1}{2}}\left(B_{\epsilon }^{n}-r_{2h}B_{\epsilon }^{n}\right){∥}^{2}\tau -2\left(d_{t}\left(u_{\epsilon }^{n}-r_{1h}u_{\epsilon }^{n}\right),\eta_{u}^{n}\right)\tau -2\left(d_{t}\left(B_{\epsilon }^{n}-r_{2h}B_{\epsilon }^{n}\right),\eta_{B}^{n}\right)\tau \\ & \;+2d\left(\eta_{u}^{n},p_{\epsilon }^{n}-\rho p_{\epsilon }^{n}\right)\tau +\frac{\epsilon }{\nu }{∥p_{\epsilon }^{n}-\rho p_{\epsilon }^{n}∥}^{2}\tau . \end{array}$$Combining (Equation 7)–(Equation 12) and Lemma 3, we obtain
$$\begin{array}{cc} & 2|b\left(u_{\epsilon }^{n-1}-u_{\epsilon h}^{n-1},u_{\epsilon }^{n},\eta_{u}^{n}\right)|\leq \frac{\nu }{8}∥A_{1h}^{\frac{1}{2}}\eta_{u}^{n}{∥}^{2}+c∥u_{\epsilon }^{n}{∥}_{2}^{2}{∥u_{\epsilon }^{n-1}-u_{\epsilon h}^{n-1}∥}^{2},\\ & 2|b\left(u_{\epsilon h}^{n-1},u_{\epsilon }^{n}-r_{1h}u_{\epsilon }^{n},\eta_{u}^{n}\right)|\leq \frac{\nu }{8}∥A_{1h}^{\frac{1}{2}}\eta_{u}^{n}{∥}^{2}+c∥u_{\epsilon h}^{n-1}{∥}_{1,2}^{2}{∥\nabla \left(u_{\epsilon }^{n}-r_{1h}u_{\epsilon }^{n}\right)∥}^{2},\\ & 2S|\left(\left(B_{\epsilon }^{n-1}-B_{\epsilon h}^{n-1}\right)\times \nabla \times B_{\epsilon }^{n},\eta_{u}^{n}\right)|\leq \frac{\nu }{8}∥A_{1h}^{\frac{1}{2}}\eta_{u}^{n}{∥}^{2}+c∥B_{\epsilon }^{n}{∥}_{2}^{2}{∥B_{\epsilon }^{n-1}-B_{\epsilon h}^{n-1}∥}^{2},\\ & 2S|\left(B_{\epsilon h}^{n-1}\times \nabla \times \left(B_{\epsilon }^{n}-r_{2h}B_{\epsilon }^{n}\right),\eta_{u}^{n}\right)|\leq \frac{\nu }{8}∥A_{1h}^{\frac{1}{2}}\eta_{u}^{n}{∥}^{2}+c∥B_{\epsilon h}^{n-1}{∥}_{1,2}^{2}{∥\nabla \left(B_{\epsilon }^{n}-r_{2h}B_{\epsilon }^{n}\right)∥}^{2},\\ & 2S|\left(u_{\epsilon }^{n}\times \left(B_{\epsilon }^{n-1}-B_{\epsilon h}^{n-1}\right),\nabla \times \eta_{B}^{n}\right)|\leq \frac{s\mu }{8}∥A_{2h}^{\frac{1}{2}}\eta_{B}^{n}{∥}^{2}+c∥u_{\epsilon }^{n}{∥}_{2}^{2}{∥B_{\epsilon }^{n-1}-B_{\epsilon h}^{n-1}∥}^{2},\\ & 2S|\left(\left(u_{\epsilon }^{n}-r_{1h}u_{\epsilon }^{n}\right)\times B_{\epsilon }^{n-1},\nabla \times \eta_{B}^{n}\right)|\leq \frac{s\mu }{8}∥A_{2h}^{\frac{1}{2}}\eta_{B}^{n}{∥}^{2}+c∥\nabla B_{\epsilon }^{n-1}{∥}^{2}{∥\nabla \left(u_{\epsilon }^{n}-r_{1h}u_{\epsilon }^{n}\right)∥}^{2},\\ & 2S|(\left(u_{\epsilon }^{n}-r_{1h}u_{\epsilon }^{n}\right)\times \left(B_{\epsilon }^{n-1}-B_{\epsilon h}^{n-1}\right),\nabla \times \eta_{B}^{n})|\\ & \;\leq \frac{s\mu }{8}∥A_{2h}^{\frac{1}{2}}\eta_{B}^{n}{∥}^{2}+c(∥B_{\epsilon }^{n-1}{∥}_{1}^{2}+∥B_{\epsilon h}^{n-1}{∥}_{1,2}^{2})∥\nabla \left(u_{\epsilon }^{n}-r_{1h}u_{\epsilon }^{n}\right){∥}^{2},\\ & 2|\left(d_{t}\left(u_{\epsilon }^{n}-r_{1h}u_{\epsilon }^{n}\right),\eta_{u}^{n}\right)\leq \frac{\nu }{8}∥A_{1h}^{\frac{1}{2}}\eta_{u}^{n}{∥}^{2}+c{∥d_{t}\left(u_{\epsilon }^{n}-r_{1h}u_{\epsilon }^{n}\right)∥}^{2},\\ & 2|\left(d_{t}\left(B_{\epsilon }^{n}-r_{2h}B_{\epsilon }^{n}\right),\eta_{B}^{n}\right)|\leq \frac{s\mu }{8}∥A_{2h}^{\frac{1}{2}}\eta_{B}^{n}{∥}^{2}+c{∥d_{t}\left(B_{\epsilon }^{n}-r_{2h}B_{\epsilon }^{n}\right)∥}^{2},\\ & 2|d\left(\eta_{u}^{n},p_{\epsilon }^{n}-\rho p_{\epsilon }^{n}\right)|\leq \frac{\nu }{8}∥A_{1h}^{\frac{1}{2}}\eta_{u}^{n}{∥}^{2}+c{∥p_{\epsilon }^{n}-\rho p_{\epsilon }^{n}∥}^{2}. \end{array}$$Combining the above inequalities with (Equation 45), and using (Equation 40) and (Equation 41), we can derive
$$\begin{array}{cc} & ∥\eta_{u}^{n}{∥}^{2}-∥\eta_{u}^{n-1}{∥}^{2}+S(∥\eta_{B}^{n}{∥}^{2}-∥\eta_{B}^{n-1}{∥}^{2})+∥d_{t}\eta_{u}^{n}{∥}^{2}\tau^{2}+S∥d_{t}\eta_{B}^{n}{∥}^{2}\tau^{2}+\nu ∥A_{1h}^{\frac{1}{2}}\left(u_{\epsilon }^{n}-u_{\epsilon h}^{n}\right){∥}^{2}\tau \\ & \;+S\mu ∥A_{2h}^{\frac{1}{2}}\left(B_{\epsilon }^{n}-B_{\epsilon h}^{n}\right){∥}^{2}\tau \leq ch^{2}(∥u_{\epsilon h}^{n-1}{∥}_{1,2}^{2}+∥B_{\epsilon h}^{n-1}{∥}_{1,2}^{2}+∥\nabla B_{\epsilon }^{n-1}{∥}^{2})(∥u_{\epsilon }^{n}{∥}_{2}^{2}+∥B_{\epsilon }^{n}{∥}_{2}^{2})\tau \\ & \;+d_{n-1}(∥u_{\epsilon }^{n-1}-u_{\epsilon h}^{n-1}{∥}^{2}+∥B_{\epsilon }^{n-1}-B_{\epsilon h}^{n-1}{∥}^{2})\tau +ch^{2}(∥u_{\epsilon }^{n}{∥}_{2}^{2}+∥B_{\epsilon }^{n}{∥}_{2}^{2}+∥d_{t}u_{\epsilon }^{n}{∥}_{1}^{2}+∥d_{t}B_{\epsilon }^{n}{∥}_{1}^{2}+∥p_{\epsilon }^{n}{∥}_{1}^{2})\tau , \end{array}$$
where $d_{n-1}=∥u_{\epsilon }^{n}{∥}^{2}+{∥B_{\epsilon }^{n}∥}^{2}$. Summing this inequality from n=1 to *m*, and using (Equation 40) and (Equation 41), Theorem 7, and Lemmas 4 and 2, we have
$$\begin{array}{cc} & ∥u_{\epsilon }^{m}-u_{\epsilon h}^{m}{∥}^{2}-S∥B_{\epsilon }^{m}-B_{\epsilon h}^{m}{∥}^{2}+\tau^{2}\sum_{n=1}^{m}(∥d_{t}\eta_{u}^{n}{∥}^{2}+S∥d_{t}\eta_{B}^{n}{∥}^{2})\\ & \;+\tau \sum_{n=1}^{m}(\nu ∥A_{1h}^{\frac{1}{2}}\left(u_{\epsilon }^{n}-u_{\epsilon h}^{n}\right){∥}^{2}+S\mu ∥A_{2h}^{\frac{1}{2}}\left(B_{\epsilon }^{n}-B_{\epsilon h}^{n}\right){∥}^{2})\\ & \;\leq \tau \sum_{n=0}^{m-1}d_{n}(∥u_{\epsilon }^{n}-u_{\epsilon h}^{n}{∥}^{2}+∥B_{\epsilon }^{n}-B_{\epsilon h}^{n}{∥}^{2})+Ch^{2}\leq Ch^{2}, \end{array}$$
for all 1≤n≤N. The proof is thus complete. □

**Theorem** **9.**
*Under Assumptions 1–4, we have*

(46)
$$\begin{array}{ccc} & & \sigma^{\frac{1}{2}}\left(t_{m}\right)(\sqrt{\nu }∥u_{\epsilon }^{m}-u_{\epsilon h}^{m}{∥}_{1}+\sqrt{\mu }∥B_{\epsilon }^{m}-B_{\epsilon h}^{m}{∥}_{1})\\ & & \;+\tau^{\frac{1}{2}}\sum_{n=1}^{m}\sigma^{\frac{1}{2}}\left(t_{n}\right)(∥d_{t}\left(u_{\epsilon }^{n}-u_{\epsilon h}^{n}\right)∥+∥d_{t}\left(B_{\epsilon }^{n}-B_{\epsilon h}^{n}\right)∥)\leq Ch, \end{array}$$


(47)
$$\tau^{\frac{1}{2}}\sum_{n=1}^{m}\sigma^{\frac{1}{2}}\left(t_{n}\right)∥p_{\epsilon }^{n}-p_{\epsilon h}^{n}∥\leq Ch.$$



**Proof.** To obtain the error estimates of the PFEM, we give the Galerkin projector $R_{1h}:\left(X,M\right)\to X_{h}$, $Q_{h}:\left(X,M\right)\to M_{h}$, which satisfies (cf. [22])
$$\begin{array}{cc} \left(\nabla \left(R_{1h}\left(u,p\right)-u\right),\nabla v_{h}\right) & -\left(\nabla \cdot v_{h},Q_{h}\left(u,p\right)-p\right)+\left(\nabla \cdot \left(R_{1h}\left(u,p\right)-u\right),q_{h}\right)\\ & +\frac{\epsilon }{\nu }\left(Q_{h}\left(u,p\right)-p,q_{h}\right)=0,\;\forall \left(v_{h},q_{h}\right)\in \left(X_{h},M_{h}\right), \end{array}$$
for all (u,p)∈(X,M) with $\nabla \cdot u+\frac{\epsilon }{\nu }p=0$. In addition, $R_{2h}:W\to W_{h}$ is the $H^{1}$-orthogonal projector defined by (cf. [4])
$$\begin{array}{c} \left(\nabla \times \left(R_{2h}B-B\right),\nabla \times C_{h}\right)+\left(\nabla \cdot \left(R_{2h}B-B\right),\nabla \cdot C_{h}\right)=0,\;\forall C_{h}\in W_{h}, \end{array}$$
for all B∈W. Due to the properties of $R_{1h}\left(u,p\right)$, $Q_{h}\left(u,p\right)$ and $R_{2h}$, we have
(48)$$∥u-R_{1h}\left(u,p\right)∥+h∥\nabla \left(u-R_{1h}\left(u,p\right)\right)∥+h∥p-Q_{h}\left(u,p\right)∥\leq Ch^{i}{(∥u∥}_{i}+{∥p∥}_{i-1}),\;i=1,2,$$
and
(49)$$∥R_{2h}B-B∥+h∥\nabla \left(R_{2h}B-B\right)∥\leq Ch^{2}{∥B∥}_{2}.$$Letting $\xi_{u}^{n}=R_{1h}\left(u_{\epsilon }^{n},p_{\epsilon }^{n}\right)-u_{\epsilon h}^{n}$, $\xi_{p}^{n}=Q_{h}\left(u_{\epsilon }^{n},p_{\epsilon }^{n}\right)-p_{\epsilon h}^{n}$, and $\xi_{B}^{n}=R_{2h}B_{\epsilon }^{n}-B_{\epsilon h}^{n}$, we derive from (Equation 43) and (Equation 44) that
(50)$$\begin{array}{cc} & \left(d_{t}\left(u_{\epsilon }^{n}-u_{\epsilon h}^{n}\right),v_{h}\right)+\nu \left(A_{1h}^{\frac{1}{2}}\xi_{u}^{n},A_{1h}^{\frac{1}{2}}v_{h}\right)+b\left(\xi_{u}^{n-1},u_{\epsilon }^{n},v_{h}\right)+b\left(u_{\epsilon h}^{n-1},\xi_{u}^{n},v_{h}\right)-\left(\nabla \cdot v_{h},\xi_{p}^{n}\right)\\ & \;+\left(\nabla \cdot d_{t}\xi_{u}^{n},q_{h}\right)+\frac{\epsilon }{\nu }\left(d_{t}\xi_{p}^{n},q_{h}\right)+S\left(\xi_{B}^{n}\times \nabla \times B_{\epsilon }^{n},v_{h}\right)+S\left(B_{\epsilon h}^{n}\times \nabla \times \xi_{B}^{n},v_{h}\right)\\ & \;=b\left(R_{1h}\left(u_{\epsilon }^{n-1},p_{\epsilon }^{n-1}\right)-u_{\epsilon }^{n-1},u_{\epsilon }^{n},v_{h}\right)+b\left(u_{\epsilon h}^{n-1},R_{1h}\left(u_{\epsilon }^{n},p_{\epsilon }^{n}\right)-u_{\epsilon }^{n},v_{h}\right)\\ & \;+S\left(R_{2h}B_{\epsilon }^{n-1}-B_{\epsilon }^{n-1}\times \nabla \times B_{\epsilon }^{n},v_{h}\right)+S\left(B_{\epsilon h}^{n-1}\times \nabla \times \left(R_{2h}B_{\epsilon }^{n}-B_{\epsilon }^{n}\right),v_{h}\right), \end{array}$$
(51)$$\begin{array}{cc} & \left(d_{t}\left(B_{\epsilon }^{n}-B_{\epsilon h}^{n}\right),C_{h}\right)+\mu \left(A_{2h}^{\frac{1}{2}}\xi_{B}^{n},A_{2h}^{\frac{1}{2}}C_{h}\right)-\left(\xi_{u}^{n}\times B_{\epsilon }^{n-1},\nabla \times C_{h}\right)-\left(u_{\epsilon h}^{n}\times \xi_{B}^{n-1},\nabla \times C_{h}\right)\\ & \;=-\left(\left(R_{1h}\left(u_{\epsilon }^{n},p_{\epsilon }^{n}\right)-u_{\epsilon }^{n}\right)\times B_{\epsilon h}^{n-1},\nabla \times C_{h}\right)-\left(u_{\epsilon h}^{n}\times \left(R_{2h}B_{\epsilon }^{n-1}-B_{\epsilon }^{n-1}\right),\nabla \times C_{h}\right). \end{array}$$Taking $v_{h}=2d_{t}\xi_{u}^{n}\tau$ and $q_{h}=2\xi_{p}^{n}\tau$ in (Equation 50) and $C_{h}=2d_{t}\xi_{B}^{n}\tau$ in (Equation 51), we obtain
(52)$$\begin{array}{cc} & 2∥d_{t}\xi_{u}^{n}{∥}^{2}\tau +2∥d_{t}\xi_{B}^{n}{∥}^{2}\tau +\nu ∥A_{1h}^{\frac{1}{2}}\xi_{u}^{n}{∥}^{2}+\mu ∥A_{2h}^{\frac{1}{2}}\xi_{B}^{n}{∥}^{2}-\nu ∥A_{1h}^{\frac{1}{2}}\xi_{u}^{n-1}{∥}^{2}-\mu ∥A_{2h}^{\frac{1}{2}}\xi_{B}^{n-1}{∥}^{2}\\ & \;+\frac{\epsilon }{\nu }(∥\xi_{p}^{n}{∥}^{2}-∥\xi_{p}^{n-1}{∥}^{2})+2b\left(u_{\epsilon }^{n-1}-u_{\epsilon h}^{n-1},u_{\epsilon }^{n},d_{t}\xi_{u}^{n}\right)\tau +2b\left(u_{\epsilon h}^{n-1},u_{\epsilon }^{n}-u_{\epsilon h}^{n},d_{t}\xi_{u}^{n}\right)\tau \\ & \;+2S\left(\left(B_{\epsilon }^{n-1}-B_{\epsilon h}^{n-1}\right)\times \nabla \times B_{\epsilon }^{n-1},d_{t}\xi_{u}^{n}\right)\tau +2S\left(B_{\epsilon h}^{n-1}\times \nabla \times \left(B_{\epsilon }^{n}-B_{\epsilon h}^{n}\right),d_{t}\xi_{u}^{n}\right)\tau \\ & \;-2\left(\left(u_{\epsilon }^{n}-u_{\epsilon h}^{n}\right)\times B_{\epsilon }^{n-1},\nabla \times d_{t}\xi_{B}^{n}\right)\tau -2\left(u_{\epsilon h}^{n}\times \left(B_{\epsilon }^{n-1}-B_{\epsilon h}^{n-1}\right),\nabla \times d_{t}\xi_{B}^{n}\right)\tau \\ & \;\leq 2\left(d_{t}\left(R_{1h}\left(u_{\epsilon }^{n},p_{\epsilon }^{n}\right)-u_{\epsilon }^{n}\right),d_{t}\xi_{u}^{n}\right)\tau +2\left(d_{t}\left(R_{2h}B_{\epsilon }^{n}-B_{\epsilon }^{n}\right),d_{t}\xi_{B}^{n}\right)\tau . \end{array}$$By using (Equation 7)–(Equation 12), (Equation 48) and (Equation 49), we obtain
$$\begin{array}{cc} & 2|b\left(u_{\epsilon }^{n-1}-u_{\epsilon h}^{n-1},u_{\epsilon }^{n},d_{t}\xi_{u}^{n}\right)|+2|b\left(u_{\epsilon h}^{n-1},u_{\epsilon }^{n}-u_{\epsilon h}^{n},d_{t}\xi_{u}^{n}\right)|\\ & \;\leq 2|b\left(u_{\epsilon }^{n-1}-u_{\epsilon h}^{n-1},u_{\epsilon }^{n},d_{t}\xi_{u}^{n}\right)|+2|b\left(u_{\epsilon }^{n-1},u_{\epsilon }^{n}-u_{\epsilon h}^{n},d_{t}\xi_{u}^{n}\right)|+2|b\left(u_{\epsilon }^{n-1}-u_{\epsilon h}^{n-1},u_{\epsilon }^{n}-u_{\epsilon h}^{n},d_{t}\xi_{u}^{n}\right)|\\ & \;\leq \frac{1}{8}∥d_{t}\xi_{u}^{n}{∥}^{2}+c∥u_{\epsilon }^{n}{∥}_{2}^{2}∥\nabla \left(u_{\epsilon }^{n-1}-u_{\epsilon h}^{n-1}\right){∥}^{2}+c∥u_{\epsilon }^{n-1}{∥}_{2}^{2}{∥\nabla \left(u_{\epsilon }^{n}-u_{\epsilon h}^{n}\right)∥}^{2}\\ & \;+ch^{-2}∥\nabla \left(u_{\epsilon }^{n-1}-u_{\epsilon h}^{n-1}\right){∥}^{2}{∥\nabla \left(u_{\epsilon }^{n}-u_{\epsilon h}^{n}\right)∥}^{2},\\ & 2S|\left(\left(B_{\epsilon }^{n-1}-B_{\epsilon h}^{n-1}\right)\times \nabla \times B_{\epsilon }^{n-1},d_{t}\xi_{u}^{n}\right)\left|+2S\right|\left(B_{\epsilon h}^{n-1}\times \nabla \times \left(B_{\epsilon }^{n}-B_{\epsilon h}^{n}\right),d_{t}\xi_{u}^{n}\right)|\\ & \;\leq \frac{1}{8}∥d_{t}\xi_{u}{∥}^{2}+c∥B_{\epsilon }^{n}{∥}_{2}^{2}∥\nabla \left(B_{\epsilon }^{n-1}-B_{\epsilon h}^{n-1}\right){∥}^{2}+c∥B_{\epsilon }^{n-1}{∥}_{2}^{2}{∥\nabla \left(B_{\epsilon }^{n}-B_{\epsilon h}^{n}\right)∥}^{2}\\ & \;+ch^{-2}∥\nabla \left(B_{\epsilon }^{n-1}-B_{\epsilon h}^{n-1}\right){∥}^{2}{∥\nabla \left(B_{\epsilon }^{n}-B_{\epsilon h}^{n}\right)∥}^{2},\\ & 2|\left(\left(u_{\epsilon }^{n}-u_{\epsilon h}^{n}\right)\times B_{\epsilon }^{n-1},\nabla \times d_{t}\xi_{B}^{n}\right)\left|+2\right|\left(u_{\epsilon h}^{n}\times \left(B_{\epsilon }^{n-1}-B_{\epsilon h}^{n-1}\right),\nabla \times d_{t}\xi_{B}^{n}\right)|\\ & \;\leq \frac{1}{8}∥d_{t}\xi_{B}^{n}{∥}^{2}+c∥B_{\epsilon }^{n-1}{∥}_{2}^{2}∥\nabla \left(u_{\epsilon }^{n}-u_{\epsilon h}^{n}\right){∥}^{2}+c∥u_{\epsilon }^{n}{∥}_{2}^{2}{∥\nabla \left(B_{\epsilon }^{n-1}-B_{\epsilon h}^{n-1}\right)∥}^{2}\\ & \;+ch^{-2}∥\nabla \left(u_{\epsilon }^{n}-u_{\epsilon h}^{n}\right){∥}^{2}{∥\nabla \left(B_{\epsilon }^{n-1}-B_{\epsilon h}^{n-1}\right)∥}^{2},\\ & 2|\left(d_{t}\left(R_{1h}\left(u_{\epsilon }^{n},p_{\epsilon }^{n}\right)-u_{\epsilon }^{n}\right),d_{t}\xi_{u}^{n}\right)\left|+2\right|\left(d_{t}\left(R_{2h}B_{\epsilon }^{n}-B_{\epsilon }^{n}\right),d_{t}\xi_{B}^{n}\right)|\\ & \;\leq \frac{1}{8}∥d_{t}\xi_{u}^{n}{∥}^{2}+\frac{1}{8}∥d_{t}\xi_{B}^{n}{∥}^{2}+ch^{4}(∥d_{t}u_{\epsilon }^{n}{∥}_{2}^{2}+∥d_{t}B_{\epsilon }^{n}{∥}_{2}^{2}+∥d_{t}p_{\epsilon }^{n}{∥}_{1}^{2}). \end{array}$$Combining the above inequalities with (Equation 52), and using Theorems 4 and 8, we can derive
$$\begin{array}{cc} & ∥d_{t}\xi_{u}^{n}{∥}^{2}\tau +∥d_{t}\xi_{B}^{n}{∥}^{2}\tau +\nu (∥A_{1h}^{\frac{1}{2}}\xi_{u}^{n}{∥}^{2}-∥A_{1h}^{\frac{1}{2}}\xi_{u}^{n-1}{∥}^{2})+\mu (∥A_{2h}^{\frac{1}{2}}\xi_{B}^{n}{∥}^{2}-∥A_{2h}^{\frac{1}{2}}\xi_{B}^{n-1}{∥}^{2})+\frac{\epsilon }{\nu }(∥\xi_{p}^{n}{∥}^{2}-∥\xi_{p}^{n-1}{∥}^{2})\\ & \;\leq c(∥\nabla \left(u_{\epsilon }^{n-1}-u_{\epsilon h}^{n-1}\right){∥}^{2}+∥\nabla \left(u_{\epsilon }^{n}-u_{\epsilon h}^{n}\right){∥}^{2}+∥\nabla \left(B_{\epsilon }^{n-1}-B_{\epsilon h}^{n-1}\right){∥}^{2}+∥\nabla \left(B_{\epsilon }^{n}-B_{\epsilon h}^{n}\right){∥}^{2})\tau \\ & \;+ch^{4}(∥d_{t}u_{\epsilon }^{n}{∥}_{2}^{2}+∥d_{t}B_{\epsilon }^{n}{∥}_{2}^{2}+∥d_{t}p_{\epsilon }^{n}{∥}_{1}^{2})\tau +ch^{-2}d_{n-1}(∥\nabla \xi_{u}^{n-1}{∥}^{2}+∥\nabla \xi_{B}^{n-1}{∥}^{2})\tau , \end{array}$$
where $d_{n-1}=ch^{-2}(∥\nabla \left(u_{\epsilon }^{n}-u_{\epsilon h}^{n}\right){∥}^{2}+∥\nabla \left(B_{\epsilon }^{n}-B_{\epsilon h}^{n}\right){∥}^{2})$. Multiplying this inequality by $\sigma (t_{n})$ and taking the sum with respect to *n* from 1 to *m*, thanks to Theorems 4 and 8 and Lemma 2, we obtain
(53)$$\begin{array}{cc} & \sigma \left(t_{m}\right)(\nu ∥A_{1h}^{\frac{1}{2}}\xi_{u}^{n}{∥}^{2}+\mu (∥A_{2h}^{\frac{1}{2}}\xi_{B}^{n}{∥}^{2})+\tau \sum_{n=1}^{m}\sigma \left(t_{n}\right)(∥d_{t}\xi_{u}^{n}{∥}^{2}+∥d_{t}\xi_{B}^{n}{∥}^{2})\\ & \;\leq \tau \sum_{n=0}^{m-1}d_{n}(∥\nabla \xi_{u}^{n}{∥}^{2}+∥\nabla \xi_{B}^{n}{∥}^{2})+Ch^{2}\leq Ch^{2}. \end{array}$$Moreover, by using (Equation 48) and (Equation 49) and Theorem 4, we obtain
(54)$$\begin{array}{c} ∥\nabla \left(R_{1h}\left(u_{\epsilon }^{n},p_{\epsilon }^{n}\right)-u_{\epsilon }^{n}\right){∥}^{2}+∥\nabla \left(R_{2h}B_{\epsilon }^{n}-B_{\epsilon }^{n}\right){∥}^{2}\leq ch^{2}(∥u_{\epsilon }^{n}{∥}_{2}^{2}+∥p_{\epsilon }^{n}{∥}_{1}^{2}+∥B_{\epsilon }^{n}{∥}_{2}^{2})\leq Ch^{2}. \end{array}$$Hence, by combining (Equation 53) with (Equation 54), we obtain (Equation 46).Next, using (Equation 7)–(Equation 12), (Equation 33) and (Equation 43), we derive
$$\begin{array}{ccc} & & \beta ∥\xi_{p}^{n}∥\leq c∥d_{t}\left(u_{\epsilon }^{n}-u_{\epsilon h}^{n}\right)∥+\nu ∥\nabla \left(u_{\epsilon }^{n}-u_{\epsilon h}^{n}\right)∥+c∥\nabla \left(u_{\epsilon }^{n-1}-u_{\epsilon h}^{n-1}\right)∥\;∥\nabla u_{\epsilon }^{n}∥\\ & & \;+c(∥\nabla u_{\epsilon }^{n-1}∥+∥\nabla \left(u_{\epsilon }^{n-1}-u_{\epsilon h}^{n-1}\right)∥)∥\nabla \left(u_{\epsilon }^{n}-u_{\epsilon h}^{n}\right)∥+c∥p_{\epsilon }^{n}-Q_{h}\left(u_{\epsilon }^{n},p_{\epsilon }^{n}\right)∥\\ & & \;+c∥\nabla \left(B_{\epsilon }^{n-1}-B_{\epsilon h}^{n-1}\right)∥\;∥\nabla B_{\epsilon }^{n}∥+c(∥\nabla B_{\epsilon }^{n-1}∥+∥\nabla \left(B_{\epsilon }^{n-1}-B_{\epsilon h}^{n-1}\right)∥)∥\nabla \left(B_{\epsilon }^{n}-B_{\epsilon h}^{n}\right)∥. \end{array}$$Multiplying this inequality by $\sigma (t_{n})$ and taking the sum with respect to *n* from 1 to *m*, due to Theorem 8 and (Equation 46), we have
(55)$$\begin{array}{ccc} & & \tau \sum_{n=1}^{m}\sigma \left(t_{n}\right){∥\xi_{p}^{n}∥}^{2}\leq Ch^{2}. \end{array}$$Using (Equation 55) and (Equation 48) and (Equation 43), we obtain (Equation 47). Thus, this proof is thus complete. □

Next, we make the $L^{2}$-error estimates. Taking $v_{h}=2A_{1h}^{-1}\eta_{u}^{n}\tau$ and $q_{h}=0$ in (Equation 43) and $C_{h}=2A_{2h}^{-1}\eta_{B}^{n}\tau$ in (Equation 44), we obtain
(56)$$\begin{array}{cc} & ∥\eta_{u}^{n}{∥}_{-1,2}^{2}+∥\eta_{B}^{n}{∥}_{-1,2}^{2}-(∥\eta_{u}^{n-1}{∥}_{-1,2}^{2}+∥\eta_{B}^{n-1}{∥}_{-1,2}^{2})+∥d_{t}\eta_{u}^{n}{∥}_{-1,2}^{2}\tau^{2}+∥d_{t}\eta_{B}^{n}{∥}_{-1,2}^{2}\tau^{2}+\mu ∥\eta_{B}^{n}{∥}^{2}\tau +\nu {∥\eta_{u}^{n}∥}^{2}\tau \\ & \;+\nu ∥u_{\epsilon }^{n}-u_{\epsilon h}^{n}{∥}^{2}\tau +\mu {∥B_{\epsilon }^{n}-B_{\epsilon h}^{n}∥}^{2}\tau +2b\left(u_{\epsilon }^{n-1}-u_{\epsilon h}^{n-1},u_{\epsilon }^{n},A_{1h}^{-1}\eta_{u}^{n}\right)\tau \\ & \;+2b\left(u_{\epsilon h}^{n-1},u_{\epsilon }^{n}-u_{\epsilon h}^{n},A_{1h}^{-1}\eta_{u}^{n}\right)\tau +2S\left(\left(B_{\epsilon }^{n-1}-B_{\epsilon h}^{n-1}\right)\times \nabla \times B_{\epsilon }^{n-1},A_{1h}^{-1}\eta_{u}^{n}\right)\tau \\ & \;+2S\left(B_{\epsilon h}^{n-1}\times \nabla \times \left(B_{\epsilon }^{n}-B_{\epsilon h}^{n}\right),A_{1h}^{-1}\eta_{u}^{n}\right)\tau -2\left(\left(u_{\epsilon }^{n}-u_{\epsilon h}^{n}\right)\times B_{\epsilon }^{n-1},\nabla \times A_{2h}^{-1}\eta_{B}^{n}\right)\tau \\ & \;-2\left(u_{\epsilon h}^{n}\times \left(B_{\epsilon }^{n-1}-B_{\epsilon h}^{n-1}\right),\nabla \times A_{2h}^{-1}\eta_{B}^{n}\right)\tau =-2\left(d_{t}\left(u_{\epsilon }^{n}-r_{1h}u_{\epsilon }^{n}\right),A_{1h}^{-1}\eta_{u}^{n}\right)\tau \\ & \;-2\left(d_{t}\left(B_{\epsilon }^{n}-r_{2h}B_{\epsilon }^{n}\right),A_{2h}^{-1}\eta_{B}^{n}\right)\tau +\nu ∥r_{1h}u_{\epsilon }^{n}-u_{\epsilon }^{n}{∥}^{2}\tau +\mu {∥r_{2h}B_{\epsilon }^{n}-B_{\epsilon }^{n}∥}^{2}\tau . \end{array}$$By using (Equation 11), (Equation 12), and Lemma 3, we have
$$\begin{array}{cc} & 2|b(u_{\epsilon }^{n-1}-u_{\epsilon h}^{n-1},u_{\epsilon }^{n},A_{1h}^{-1}\eta_{u}^{n})|\\ & \;\leq 2|b\left(u_{\epsilon }^{n-1}-r_{1h}u_{\epsilon h}^{n-1},u_{\epsilon }^{n},A_{1h}^{-1}\eta_{u}^{n}\right)|+2|b\left(\eta_{u}^{n-1},u_{\epsilon }^{n}-u_{\epsilon h}^{n},A_{1h}^{-1}\eta_{u}^{n}\right)|+2|b\left(\eta_{u}^{n-1},u_{\epsilon h}^{n},A_{1h}^{-1}\eta_{u}^{n}\right)|\\ & \;\leq \frac{\nu }{8}∥\eta_{u}^{n}{∥}^{2}+c∥u_{\epsilon }^{n}{∥}_{1}^{2}∥u_{\epsilon }^{n-1}-r_{1h}u_{\epsilon }^{n-1}{∥}^{2}+c∥u_{\epsilon h}^{n}{∥}_{2,2}^{2}∥\eta_{u}^{n-1}{∥}_{-1,2}^{2}+c∥\eta_{u}^{n-1}{∥}^{2}{∥u_{\epsilon }^{n}-u_{\epsilon h}^{n}∥}_{1}^{2},\\ & 2|b(u_{\epsilon h}^{n-1},u_{\epsilon }^{n}-u_{\epsilon h}^{n},A_{1h}^{-1}\eta_{u}^{n})|\\ & \;\leq 2|b\left(u_{\epsilon h}^{n-1},u_{\epsilon }^{n}-u_{\epsilon h}^{n},A_{1h}^{-1}\left(\eta_{u}^{n}-\eta_{u}^{n-1}\right)\right)|+2|b\left(u_{\epsilon h}^{n-1},u_{\epsilon }^{n}-r_{1h}u_{\epsilon }^{n},A_{1h}^{-1}\eta_{u}^{n-1}\right)|+2|b\left(u_{\epsilon h}^{n-1},\eta_{u}^{n},A_{1h}^{-1}\eta_{u}^{n-1}\right)|\\ & \;\leq \frac{1}{8\tau }∥\eta_{u}^{n}-\eta_{u}^{n-1}{∥}_{-1,2}^{2}+\frac{\nu }{8}(∥\eta_{u}^{n}{∥}^{2}+∥u_{\epsilon }^{n}-r_{1h}u_{\epsilon }^{n}{∥}^{2})+c\tau ∥u_{\epsilon h}^{n-1}{∥}_{1,2}^{2}∥u_{\epsilon }^{n}-u_{\epsilon h}^{n}{∥}_{1}^{2}+c∥u_{\epsilon h}^{n-1}{∥}_{2,2}^{2}{∥\eta_{u}^{n-1}∥}_{-1,2}^{2},\\ & 2S|(\left(B_{\epsilon }^{n-1}-B_{\epsilon h}^{n-1}\right)\times \nabla \times B_{\epsilon }^{n-1},A_{1h}^{-1}\eta_{u}^{n})|\\ & \;\leq \frac{\nu }{8}∥\eta_{u}^{n}{∥}^{2}+c∥B_{\epsilon }^{n-1}-r_{2h}B_{\epsilon }^{n-1}{∥}^{2}∥B_{\epsilon }^{n}{∥}_{1}^{2}+c∥\eta_{B}^{n-1}{∥}^{2}∥B_{\epsilon }^{n}-B_{\epsilon h}^{n}{∥}_{1}^{2}+c∥B_{\epsilon h}^{n}{∥}_{2,2}^{2}{∥\eta_{B}^{n-1}∥}_{-1,2}^{2},\\ & 2S|(B_{\epsilon h}^{n-1}\times \nabla \times \left(B_{\epsilon }^{n}-B_{\epsilon h}^{n}\right),A_{1h}^{-1}\eta_{u}^{n})|\\ & \;\leq \frac{1}{8\tau }∥\eta_{u}^{n}-\eta_{u}^{n-1}{∥}_{-1,2}^{2}+\frac{\mu }{8}(∥\eta_{B}^{n}{∥}^{2}+∥B_{\epsilon }^{n}-r_{2h}B_{\epsilon }^{n}{∥}^{2})+c\tau ∥B_{\epsilon h}^{n-1}{∥}_{1,2}^{2}∥B_{\epsilon }^{n}-B_{\epsilon h}^{n}{∥}_{1}^{2}+c∥B_{\epsilon h}^{n-1}{∥}_{2,2}^{2}{∥\eta_{u}^{n-1}∥}_{-1,2}^{2},\\ & 2|(\left(u_{\epsilon }^{n}-u_{\epsilon h}^{n}\right)\times B_{\epsilon }^{n-1},\nabla \times A_{2h}^{-1}\eta_{B}^{n})|\\ & \;\leq \frac{1}{8\tau }∥\eta_{B}^{n}-\eta_{B}^{n-1}{∥}_{-1,2}^{2}+c\tau ∥B_{\epsilon }^{n-1}{∥}_{1}^{2}∥B_{\epsilon }^{n}-B_{\epsilon h}^{n}{∥}_{1}^{2}+\frac{\mu }{8}(∥\eta_{u}^{n}{∥}^{2}+∥u_{\epsilon }^{n}-r_{1h}u_{\epsilon }^{n}{∥}^{2})+c∥B_{\epsilon }^{n-1}{∥}_{2}^{2}{∥\eta_{B}^{n-1}∥}_{-1,2}^{2},\\ & 2|\left(u_{\epsilon h}^{n}\times \left(B_{\epsilon }^{n-1}-B_{\epsilon h}^{n-1}\right),\nabla \times A_{2h}^{-1}\eta_{B}^{n}\right)|\leq \frac{\mu }{8}∥\eta_{B}^{n}{∥}^{2}+c∥B_{\epsilon }^{n-1}-r_{2h}B_{\epsilon }^{n-1}{∥}^{2}∥B_{\epsilon h}^{n}{∥}_{1,2}^{2}+c∥u_{\epsilon h}^{n}{∥}_{2,2}^{2}{∥\eta_{B}^{n-1}∥}_{-1,2}^{2},\\ & 2|\left(d_{t}\left(u_{\epsilon }^{n}-r_{1h}u_{\epsilon }^{n}\right),A_{1h}^{-1}\eta_{u}^{n}\right)\left|+2\right|\left(d_{t}\left(B_{\epsilon }^{n}-r_{2h}B_{\epsilon }^{n}\right),A_{2h}^{-1}\eta_{B}^{n}\right)|\\ & \;\leq \frac{1}{8}(\nu ∥\eta_{u}^{n}{∥}^{2}+\mu ∥\eta_{B}^{n}{∥}^{2})+c∥d_{t}\left(u_{\epsilon }^{n}-r_{1h}u_{\epsilon }^{n}\right){∥}^{2}+c{∥d_{t}\left(B_{\epsilon }^{n}-r_{2h}B_{\epsilon }^{n}\right)∥}^{2}. \end{array}$$Combining these inequalities with (Equation 56), using (Equation 40) and (Equation 41), we obtain
(57)$$\begin{array}{cc} & ∥\eta_{u}^{n}{∥}_{-1,2}^{2}+∥\eta_{B}^{n}{∥}_{-1,2}^{2}-(∥\eta_{u}^{n-1}{∥}_{-1,2}^{2}+∥\eta_{B}^{n-1}{∥}_{-1,2}^{2})+\mu ∥\eta_{B}^{n}{∥}^{2}\tau +\nu {∥\eta_{u}^{n}∥}^{2}\tau \\ & \;\leq d_{n-1}(∥\eta_{u}^{n-1}{∥}_{-1,2}^{2}+∥\eta_{B}^{n-1}{∥}_{-1,2}^{2})\tau +ch^{4}(∥u_{\epsilon }^{n}{∥}_{2}^{2}+∥B_{\epsilon }^{n}{∥}_{2}^{2}+∥p_{\epsilon }^{n}{∥}_{1}^{2})\tau \\ & \;+ch^{4}(∥u_{\epsilon h}^{n}{∥}_{1,2}^{2}+∥u_{\epsilon }^{n}{∥}_{1}^{2}+∥B_{\epsilon }^{n}{∥}_{1}^{2})(∥u_{\epsilon }^{n}{∥}_{2}^{2}+∥B_{\epsilon }^{n}{∥}_{2}^{2}+∥p_{\epsilon }^{n}{∥}_{1}^{2})\tau \\ & \;+c∥\eta_{u}^{n-1}{∥}^{2}∥u_{\epsilon }^{n}-u_{\epsilon h}^{n}{∥}^{2}\tau +c∥\eta_{B}^{n-1}{∥}^{2}{∥B_{\epsilon }^{n}-B_{\epsilon h}^{n}∥}_{1}^{2}\tau \\ & \;+c(∥u_{\epsilon h}^{n}{∥}^{2}+∥B_{\epsilon h}^{n-1}{∥}^{2}+∥B_{\epsilon }^{n-1}{∥}^{2})(∥u_{\epsilon }^{n}-u_{\epsilon h}^{n}{∥}_{1}^{2}+∥B_{\epsilon }^{n}-B_{\epsilon h}^{n}{∥}_{1}^{2})\tau^{2}, \end{array}$$
where $d_{n-1}=∥u_{\epsilon h}^{n}{∥}_{2,2}^{2}+∥B_{\epsilon h}^{n}{∥}_{2,2}^{2}+∥u_{\epsilon h}^{n-1}{∥}_{2,2}^{2}+∥B_{\epsilon h}^{n-1}{∥}_{2,2}^{2}+{∥B_{\epsilon }^{n-1}∥}_{2}^{2}$. In addition, we have
$$\begin{array}{c} ∥u_{\epsilon h}^{n}{∥}_{2,2}\leq ch^{-1}∥u_{\epsilon }^{n}-u_{\epsilon h}^{n}{∥}_{1}+∥u_{\epsilon }^{n}{∥}_{2},\;∥B_{\epsilon h}^{n}{∥}_{2,2}\leq ch^{-1}∥B_{\epsilon }^{n}-B_{\epsilon h}^{n}{∥}_{1}+{∥B_{\epsilon }^{n}∥}_{2}. \end{array}$$

Summing (Equation 57) from n=1 to *m*, and using Lemma 4 and Theorems 8 and 9, we have
(58)$$\begin{array}{cc} & ∥\eta_{u}^{n}{∥}_{-1,2}^{2}+∥\eta_{B}^{n}{∥}_{-1,2}^{2}+\tau \sum_{n=1}^{m}(\mu ∥\eta_{B}^{n}{∥}^{2}+\nu ∥\eta_{u}^{n}{∥}^{2})\leq \tau \sum_{n=0}^{m-1}d_{n}(∥\eta_{u}^{n}{∥}_{-1,2}^{2}+∥\eta_{B}^{n}{∥}_{-1,2}^{2})+Ch^{4}+Ch^{2}\tau . \end{array}$$Then, applying Lemma 2 to (Equation 58) and using (Equation 40) and (Equation 41), we obtain
(59)$$\begin{array}{cc} & ∥\eta_{u}^{n}{∥}_{-1,2}^{2}+∥\eta_{B}^{n}{∥}_{-1,2}^{2}+\tau \sum_{n=1}^{m}(\mu ∥u_{\epsilon }^{n}-u_{\epsilon h}^{n}{∥}^{2}+\nu ∥B_{\epsilon }^{n}-B_{\epsilon h}^{n}{∥}^{2})\leq Ch^{4}+C\tau^{2}. \end{array}$$

**Theorem** **10.**
*Under the assumption of Lemma 4, we have*

(60)
$$\begin{array}{c} \sigma^{\frac{1}{2}}\left(t_{m}\right)(∥u_{\epsilon }^{m}-u_{\epsilon h}^{m}∥+∥B_{\epsilon }^{m}-B_{\epsilon h}^{m}∥)\leq C\left(h^{2}+\tau \right). \end{array}$$



**Proof.** Taking $v_{h}=2\xi_{u}^{n}\tau$ and $q_{h}=2\xi_{p}^{n}\tau$ in (Equation 43) and $C_{h}=2\xi_{B}^{n}\tau$ in (Equation 44), and adding these equations together, we have
(61)$$\begin{array}{cc} & ∥\xi_{u}^{n}{∥}^{2}+∥\xi_{B}^{n}{∥}^{2}-(∥\xi_{u}^{n-1}{∥}^{2}+∥\xi_{B}^{n-1}{∥}^{2})+2(\nu ∥\xi_{u}^{n}{∥}_{1}^{2}+\mu ∥\xi_{B}^{n}{∥}_{1}^{2}+\frac{\nu }{\epsilon }∥\xi_{p}^{n}{∥}^{2})\tau +2b\left(u_{\epsilon }^{n-1}-u_{\epsilon h}^{n-1},u_{\epsilon }^{n},\xi_{u}^{n}\right)\tau \\ & \;+2b\left(u_{\epsilon h}^{n-1},u_{\epsilon }^{n}-u_{\epsilon h}^{n},\xi_{u}^{n}\right)\tau +2S\left(\left(B_{\epsilon }^{n-1}-B_{\epsilon h}^{n-1}\right)\times \nabla \times B_{\epsilon }^{n},\xi_{u}^{n}\right)\tau +2S\left(B_{\epsilon h}^{n-1}\times \nabla \times \left(B_{\epsilon }^{n}-B_{\epsilon h}^{n}\right),\xi_{u}^{n}\right)\tau \\ & \;-2\left(\left(u_{\epsilon }^{n}-u_{\epsilon h}^{n}\right)\times B_{\epsilon }^{n-1},\nabla \times \xi_{B}^{n}\right)\tau -2\left(u_{\epsilon h}^{n}\times \left(B_{\epsilon }^{n-1}-B_{\epsilon h}^{n-1}\right),\nabla \times \xi_{B}^{n}\right)\tau \\ & \;\leq 2\left(d_{t}\left(R_{1h}\left(u_{\epsilon }^{n},p_{\epsilon }^{n}\right)-u_{\epsilon }^{n}\right),\xi_{u}^{n}\right)\tau +2\left(d_{t}\left(R_{2h}B_{\epsilon }^{n}-B_{\epsilon }^{n}\right),\xi_{B}^{n}\right)\tau . \end{array}$$By using (Equation 11), (Equation 12), and Lemma 3, we obtain
$$\begin{array}{cc} & 2|b\left(u_{\epsilon }^{n-1}-u_{\epsilon h}^{n-1},u_{\epsilon }^{n},\xi_{u}^{n}\right)|+2|b\left(u_{\epsilon h}^{n-1},u_{\epsilon }^{n}-u_{\epsilon h}^{n},\xi_{u}^{n}\right)|\\ & \;\leq \frac{\nu }{8}∥\xi_{u}^{n}{∥}_{1}^{2}+c∥u_{\epsilon }^{n}{∥}_{2}^{2}∥u_{\epsilon }^{n-1}-u_{\epsilon h}^{n-1}{∥}^{2}+c∥u_{\epsilon }^{n-1}{∥}_{2}^{2}∥u_{\epsilon }^{n}-u_{\epsilon h}^{n}{∥}^{2}+c∥u_{\epsilon }^{n-1}-u_{\epsilon h}^{n-1}{∥}_{1}^{2}{∥u_{\epsilon }^{n}-u_{\epsilon h}^{n}∥}_{1}^{2},\\ & 2S|\left(\left(B_{\epsilon }^{n-1}-B_{\epsilon h}^{n-1}\right)\times \nabla \times B_{\epsilon }^{n},\xi_{u}^{n}\right)\left|+2S\right|\left(B_{\epsilon h}^{n-1}\times \nabla \times \left(B_{\epsilon }^{n}-B_{\epsilon h}^{n}\right),\xi_{u}^{n}\right)|\\ & \;\leq \frac{\nu }{8}∥\xi_{u}^{n}{∥}_{1}^{2}+c∥B_{\epsilon }^{n}{∥}_{2}^{2}∥B_{\epsilon }^{n-1}-B_{\epsilon h}^{n-1}{∥}^{2}+c∥B_{\epsilon }^{n-1}{∥}_{2}^{2}∥B_{\epsilon }^{n}-B_{\epsilon h}^{n}{∥}^{2}+c∥B_{\epsilon }^{n-1}-B_{\epsilon h}^{n-1}{∥}_{1}^{2}{∥B_{\epsilon }^{n}-B_{\epsilon h}^{n}∥}_{1}^{2},\\ & 2|\left(\left(u_{\epsilon }^{n}-u_{\epsilon h}^{n}\right)\times B_{\epsilon }^{n-1},\nabla \times \xi_{B}^{n}\right)\left|+2\right|\left(u_{\epsilon h}^{n}\times \left(B_{\epsilon }^{n-1}-B_{\epsilon h}^{n-1}\right),\nabla \times \xi_{B}^{n}\right)|\\ & \;\leq \frac{\mu }{8}∥\xi_{B}^{n}{∥}_{1}^{2}+c∥B_{\epsilon }^{n-1}{∥}_{2}^{2}∥u_{\epsilon }^{n}-u_{\epsilon h}^{n}{∥}^{2}+c∥u_{\epsilon }^{n}{∥}_{2}^{2}∥B_{\epsilon }^{n-1}-B_{\epsilon h}^{n-1}{∥}^{2}+c∥B_{\epsilon }^{n-1}-B_{\epsilon h}^{n-1}{∥}_{1}^{2}{∥u_{\epsilon }^{n}-u_{\epsilon h}^{n}∥}_{1}^{2},\\ & 2|\left(d_{t}\left(R_{1h}\left(u_{\epsilon }^{n},p_{\epsilon }^{n}\right)-u_{\epsilon }^{n}\right),\xi_{u}^{n}\right)\left|+2\right|\left(d_{t}\left(R_{2h}B_{\epsilon }^{n}-B_{\epsilon }^{n}\right),\xi_{B}^{n}\right)|\\ & \;\leq \frac{\nu }{8}∥\xi_{u}^{n}{∥}_{1}^{2}+\frac{\mu }{8}∥\xi_{B}^{n}{∥}_{1}^{2}+c∥d_{t}\left(R_{1h}\left(u_{\epsilon }^{n},p_{\epsilon }^{n}\right)-u_{\epsilon }^{n}\right){∥}^{2}+c∥d_{t}{(R_{2h}B_{\epsilon }^{n}-B_{\epsilon }^{n}∥}^{2}. \end{array}$$Combining these inequalities with (Equation 61) and using (Equation 40) and (Equation 41), we have
$$\begin{array}{cc} & ∥\xi_{u}^{n}{∥}^{2}+∥\xi_{B}^{n}{∥}^{2}-(∥\xi_{u}^{n-1}{∥}^{2}+∥\xi_{B}^{n-1}{∥}^{2})+(\nu ∥\xi_{u}^{n}{∥}_{1}^{2}+\mu ∥\xi_{B}^{n}{∥}_{1}^{2})\tau \\ & \;\leq c(∥u_{\epsilon }^{n}{∥}_{2}^{2}+∥B_{\epsilon }^{n}{∥}_{2}^{2})(∥u_{\epsilon }^{n-1}-u_{\epsilon h}^{n-1}∥+∥B_{\epsilon }^{n-1}-B_{\epsilon h}^{n-1}∥)\tau +ch^{4}(∥d_{t}u_{\epsilon }^{n}{∥}_{2}^{2}+∥d_{t}B_{\epsilon }^{n}{∥}_{2}^{2}+∥d_{t}p_{\epsilon }^{n}{∥}_{1}^{2})\\ & \;+c(∥u_{\epsilon }^{n-1}{∥}_{2}^{2}+∥B_{\epsilon }^{n-1}{∥}_{2}^{2})(∥u_{\epsilon }^{n}-u_{\epsilon h}^{n}∥+∥B_{\epsilon }^{n}-B_{\epsilon h}^{n}∥)\tau \\ & \;+c(∥u_{\epsilon }^{n-1}-u_{\epsilon h}^{n-1}{∥}_{1}^{2}+∥B_{\epsilon }^{n-1}-B_{\epsilon h}^{n-1}{∥}_{1}^{2})(∥u_{\epsilon }^{n}-u_{\epsilon h}^{n}{∥}_{1}^{2}+∥B_{\epsilon }^{n}-B_{\epsilon h}^{n}{∥}_{1}^{2})\tau . \end{array}$$Multiplying this inequality by $\sigma (t_{n})$ and taking the sum with respect to *n* from 1 to *m*, thanks to Theorems 4 and 8 and Lemma 2, we obtain
(62)$$\begin{array}{c} \sigma \left(t_{m}\right)(∥\xi_{u}^{m}{∥}^{2}+∥\xi_{B}^{m}{∥}^{2})+\tau \sum_{n=1}^{m}\sigma \left(t_{n}\right)(\nu ∥\xi_{u}^{n}{∥}_{1}^{2}+\mu ∥\xi_{B}^{n}{∥}_{1}^{2})\tau \leq Ch^{4}+C\tau^{2}. \end{array}$$Then, by applying (Equation 40), (Equation 41), Theorem 4, and (Equation 62), we obtain Theorem 10. □

## 6. Error Estimates

Combining Theorem 3 and the results in Section 3, Section 4 and Section 5, we obtain the following results on convergence of the fully discrete Euler semi-implicit scheme.

**Theorem** **11.**
*Under Assumptions 1–4, we have the following error estimates*

$$\begin{array}{cc} & \sigma^{\frac{1}{2}}\left(t_{m}\right)(\sqrt{\nu }∥u\left(t_{m}\right)-u_{\epsilon h}^{m}{∥}_{1}+\sqrt{\mu }∥B\left(t_{m}\right)-B_{\epsilon h}^{m}{∥}_{1})+{\left(\tau \sum_{n=1}^{m}\sigma \left(t_{n}\right){∥p\left(t_{n}\right)-p_{\epsilon h}^{n}∥}^{2}\right)}^{\frac{1}{2}}\leq C\left(h+\tau +\epsilon \right),\\ & \sigma^{\frac{1}{2}}\left(t_{m}\right)(∥u\left(t_{m}\right)-u_{\epsilon h}^{m}∥+∥B\left(t_{m}\right)-B_{\epsilon h}^{m}∥)\leq C\left(h^{2}+\tau +\epsilon \right). \end{array}$$



## 7. Numerical Example

In this part, we present two numerical tests to validate the accuracy and performance of our scheme. We use the $P_{1}^{b}/P_{1}$ element that satisfies the LBB condition for velocity and pressure (u,p), and the $P_{1}$ element for magnetic field B. The penalty parameter is selected as ϵ=τ in all the numerical tests.

### 7.1. Convergence Tests

We verify the convergence rates of the PFEM based on the Euler implicit scheme in this example. We use the computational domain ${\left[0,1\right]}^{d},d=2,3$ and set parameters ν=μ=S=1. The source terms are given by the following exact solutions:$$\left\{\begin{array}{c} u=(y\left(y-1\right)\left(2y-1\right)x^{2}{\left(x-1\right)}^{2}\cos\left(t\right),-x\left(x-1\right)\left(2x-1\right)y^{2}{\left(y-1\right)}^{2}\cos\left(t\right)),\\ p=(2y-1)(2x-1)\cos(t),\\ B=(\cos(\pi y)\sin(\pi x)\cos(t),-\cos(\pi x)\sin(\pi y)\cos(t)), \end{array}\right.$$
and
$$\left\{\begin{array}{c} u=(\left(y^{4}+z^{2}\right)\exp\left(-t\right),\left(z^{4}+x^{2}\right)\exp\left(-t\right),\left(x^{4}+y^{2}\right)\exp\left(-t\right)),\\ p=(2x-1)(2y-1)(2z-1)\exp(-t),\\ B=((\sin(y)+z)\exp(-t),(\sin(z)+x)\exp(-t),(\sin(x)+y)\exp(-t)). \end{array}\right.$$

We choose $\tau =h^{2}$ and h=1/n (n=8,16,32,64,128 in $R^{2}$ or n=4,8,12,16,20$R^{3}$). The numerical errors and the space convergence rates of the PFEM based on the semi-implicit scheme at $t_{n}=1$ s are presented in Table 1 and Table 2. We observe the first order accuracy for $H^{1}$ errors of u,B, and the second order accuracy asymptotically for $L^{2}$ errors of u,B, which are consistent with our theoretical results. These convergence rates are consistent with the expected orders. Notice that $L^{2}$ errors of *p* has a faster convergence rate than the theoretical results.

### 7.2. Two-Sided Lid-Driven Square Cavity Flow

In this example, we test the 2D/3D two-sided driven cavity flow problem (cf. [33]). The problem we study is the incompressible viscous flow in a square cavity whose top and bottom walls move in the same (parallel) or opposite (antiparallel) direction. We take the initial values $u^{0}=B^{0}=0$ and the source terms f=g=0. In the 2D case, we set a computational domain as $D={\left[0,1\right]}^{2}$. The two boundary conditions are shown below:$$\left\{\begin{array}{c} u=0,\;\mathrm{on}\;x=0,1,\\ u=(1,0),\;\mathrm{on}\;y=0,1,\\ n\times B=(1,0)\times B\;\mathrm{on}\;\Gamma , \end{array}\right.\;\left\{\begin{array}{c} u=0,\;\mathrm{on}\;x=0,1,\\ u=(1,0),\;\mathrm{on}\;y=1,\\ u=(-1,0),\;\mathrm{on}\;y=0,\\ n\times B=(1,0)\times B\;\mathrm{on}\;\Gamma . \end{array}\right.$$

We set $h=\frac{1}{60}$, $\tau =\frac{1}{3600}$. First, we consider the upper and lower walls moving in the same direction at the same speed along the *x*-axis. Figure 1 shows the velocity streamlines for the fluid Reynolds number $R_{e}=1,100,1000$, the magnetic Reynolds number $R_{m}=1$, and the coupling coefficient S=10. It can be observed that the velocity streamlines are symmetric lines parallel to these walls and pass through the center of the cavity. With the increase of the fluid Reynolds number $R_{e}$, the centers of the two symmetric vortices move to the right, and the two symmetric vortices become four symmetric vortices. Figure 2 shows the velocity streamlines for the fluid Reynolds number $R_{e}=100$, the magnetic Reynolds number $R_{m}=1$ and the coupling coefficient S=1,100,1000. With the increase of coupling coefficient *S*, the two symmetric large vortices can split into more and more small vortices.

Then, we consider the upper and lower walls moving in the opposite direction at the same speed along the *x*-axis. Figure 3 gives the velocity streamlines for the fluid Reynolds number $R_{e}=1,100,1000$, the magnetic Reynolds number $R_{m}=1$ and the coupling coefficient S=1. We find that, with the increase of the fluid Reynolds number $R_{e}$, the centers of the two symmetric vortices shift to the upper right corner and the lower left corner, respectively. Figure 4 presents the velocity streamlines for the fluid Reynolds number $R_{e}=1$, the magnetic Reynolds number $R_{m}=1$ and the coupling coefficient S=100,1000,10000. It can be observed from the figure that, with the increase of coupling coefficient *S*, the two symmetric large vortices become four small vortices.

In the 3D case, we set a calculational domain is $D={\left[0,1\right]}^{3}$. The two boundary conditions are shown below:$$\left\{\begin{array}{c} u=0,\;\mathrm{on}\;x=0,1,\\ u=(1,0,0),\;\mathrm{on}\;y=0,1,\\ n\times B=(1,0,0)\times B\;\mathrm{on}\;\Gamma , \end{array}\right.\;\left\{\begin{array}{c} u=0,\;\mathrm{on}\;x=0,1,\\ u=(1,0,0),\;\mathrm{on}\;y=1,\\ u=(-1,0,0),\;\mathrm{on}\;y=0,\\ n\times B=(1,0,0)\times B\;\mathrm{on}\;\Gamma . \end{array}\right.$$

We set $h=\frac{1}{12}$, $\tau =\frac{1}{600}$. First, we consider the top and bottom walls moving in the same direction at the same speed. Figure 5 shows the velocity streamlines at plane y=0.5 for the fluid Reynolds number $R_{e}=1,100,500$, the magnetic Reynolds number $R_{m}=1$ and the coupling coefficient S=1. We find that, with the increase of the fluid Reynolds number $R_{e}$, the centers of the two symmetric vortices move to the right. Figure 6 shows the velocity streamlines at plane y=0.5 for the fluid Reynolds number $R_{e}=10$, the magnetic Reynolds number $R_{m}=1$ and the the coupling coefficient S=1,100,500. We find that the two symmetric large vortices can split into more and more small vortices.

Then, we consider the top and bottom walls moving in the opposite direction at the same speed. Figure 7 gives the velocity streamlines at plane y=0.5 for the fluid Reynolds number $R_{e}=1,100,500$, the magnetic Reynolds number $R_{m}=1$, and the coupling coefficient S=1. We can see the centers of the two symmetric vortices shift to the upper right corner and the lower left corner, respectively. Figure 8 presents the velocity streamlines at plane y=0.5 for the fluid Reynolds number $R_{e}=1$, the magnetic Reynolds number $R_{m}=1$, and the coupling coefficient S=1,100,500. With the increase of coupling coefficient *S*, the centers of the two symmetric vortices are slightly offset.

## 8. Conclusions

We present the fully discrete PFEM for the 2D/3D unsteady MHD equations in this paper. We introduce a penalty term to decouple the MHD equations into two small equations: one is the equations of velocity and magnetic field (u,B), and the other is the equation of pressure *p*. Furthermore, we derive the error estimates for our scheme. Finally, two 2D/3D numerical experiments are given to verify the theoretical results.

## Figures and Tables

**Figure 1 entropy-24-01395-f001:**
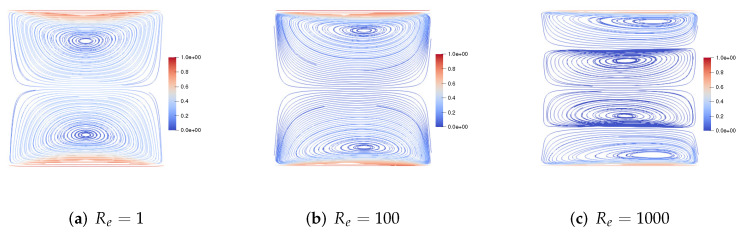
The velocity streamlines of the upper and lower walls moving in the same direction for $R_{e}=1,100,1000,R_{m}=1,S=10$.

**Figure 2 entropy-24-01395-f002:**
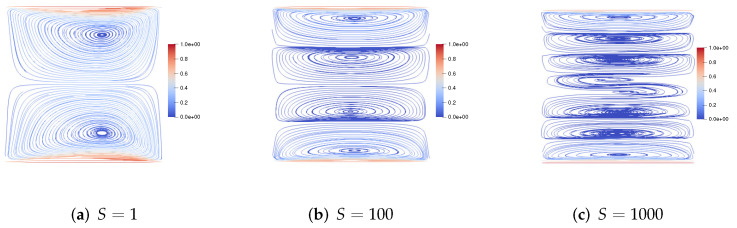
The velocity streamlines of the upper and lower walls moving in the same direction for $R_{e}=100,R_{m}=1,S=1,100,1000$.

**Figure 3 entropy-24-01395-f003:**
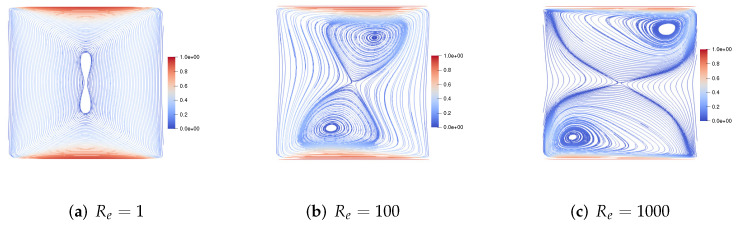
The velocity streamlines of the upper and lower walls moving in the opposite direction for $R_{e}=1,100,1000,R_{m}=1,S=1$.

**Figure 4 entropy-24-01395-f004:**
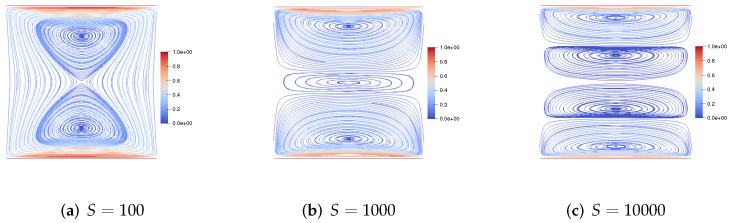
The velocity streamlines of the upper and lower walls moving in the opposite direction for $R_{e}=1,R_{m}=1,S=100,1000,10,000$.

**Figure 5 entropy-24-01395-f005:**
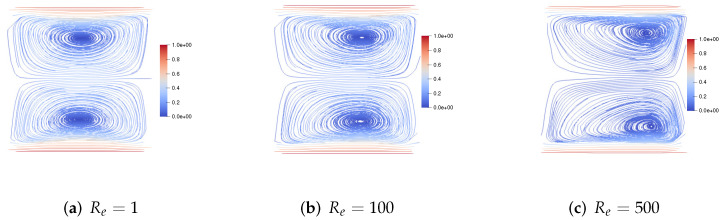
The velocity streamlines of the upper and lower walls moving in the same direction for $R_{e}=1,100,500,R_{m}=1,S=1$.

**Figure 6 entropy-24-01395-f006:**
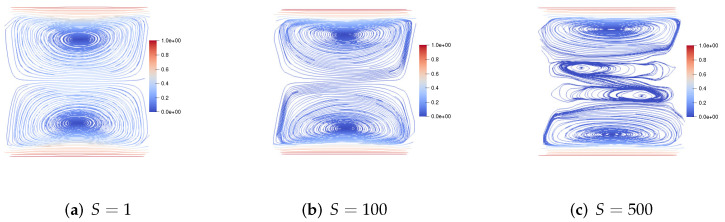
The velocity streamlines of the upper and lower walls moving in the same direction for $R_{e}=10,R_{m}=1,S=1,100,500$.

**Figure 7 entropy-24-01395-f007:**
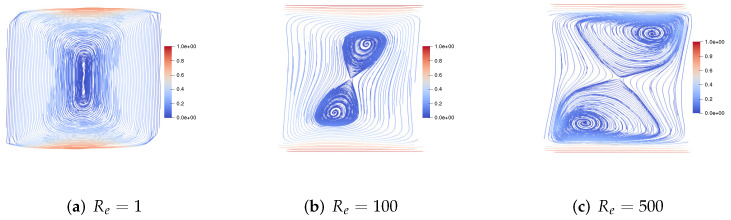
The velocity streamlines of the upper and lower walls moving in the opposite direction for $R_{e}=1,100,500,R_{m}=1,S=1$.

**Figure 8 entropy-24-01395-f008:**
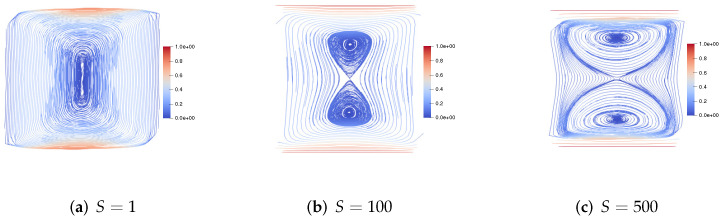
The velocity streamlines of the upper and lower walls moving in the opposite direction for $R_{e}=1,R_{m}=1,S=1,100,500$.

**Table 1 entropy-24-01395-t001:** The convergence rates of our scheme at $t_{n}=1$ s (2D).

h	$∥u-u_{\epsilon h}^{n}∥$	Ratio	$∥u-u_{\epsilon h}^{n}{∥}_{1}$	Ratio	$∥p-p_{\epsilon h}^{n}∥$	Ratio	$∥B-B_{\epsilon h}^{n}∥$	Ratio	$∥B-B_{\epsilon h}^{n}{∥}_{1}$	Ratio
1/8	2.46 × 10${}^{-4}$		5.34 × 10${}^{-3}$		5.01 × 10${}^{-3}$		1.39 × 10${}^{-2}$		3.30 × 10${}^{-1}$	
1/16	6.18 × 10${}^{-5}$	2.00	2.59 × 10${}^{-3}$	1.04	1.43 × 10${}^{-3}$	1.81	3.55 × 10${}^{-3}$	1.97	1.66 × 10${}^{-1}$	0.99
1/32	1.53 × 10${}^{-5}$	2.01	1.28 × 10${}^{-3}$	1.02	4.29 × 10${}^{-4}$	1.74	8.92 × 10${}^{-4}$	1.99	8.33 × 10${}^{-2}$	1.00
1/64	3.80 × 10${}^{-6}$	2.01	6.34 × 10${}^{-4}$	1.01	1.37 × 10${}^{-4}$	1.65	2.23 × 10${}^{-4}$	2.00	4.17 × 10${}^{-2}$	1.00
1/128	9.45 × 10${}^{-6}$	2.01	3.16 × 10${}^{-4}$	1.00	4.56 × 10${}^{-5}$	1.58	5.59 × 10${}^{-5}$	2.00	2.08 × 10${}^{-2}$	1.00

**Table 2 entropy-24-01395-t002:** The convergence rates of our scheme at $t_{n}=1$ s (3D).

h	$∥u-u_{\epsilon h}^{n}∥$	Ratio	$∥u-u_{\epsilon h}^{n}{∥}_{1}$	Ratio	$∥p-p_{\epsilon h}^{n}∥$	Ratio	$∥B-B_{\epsilon h}^{n}∥$	Ratio	$∥B-B_{\epsilon h}^{n}{∥}_{1}$	Ratio
1/4	2.41 × 10${}^{-2}$		2.49 × 10${}^{-1}$		9.05 × 10${}^{-2}$		1.53 × 10${}^{-3}$		2.44 × 10${}^{-2}$	
1/8	6.11 × 10${}^{-3}$	1.98	1.26 × 10${}^{-1}$	0.99	2.95 × 10${}^{-2}$	1.61	3.60 × 10${}^{-4}$	2.08	1.21 × 10${}^{-2}$	1.01
1/12	2.72 × 10${}^{-3}$	2.00	8.38 × 10${}^{-2}$	1.00	1.46 × 10${}^{-2}$	1.73	1.58 × 10${}^{-4}$	2.03	8.02 × 10${}^{-3}$	1.01
1/16	1.53 × 10${}^{-3}$	2.00	6.29 × 10${}^{-2}$	1.00	8.84 × 10${}^{-3}$	1.75	8.83 × 10${}^{-5}$	2.02	6.01 × 10${}^{-3}$	1.00
1/20	9.80 × 10${}^{-4}$	2.00	5.03 × 10${}^{-2}$	1.00	6.00 × 10${}^{-3}$	1.74	5.64 × 10${}^{-5}$	2.01	4.81 × 10${}^{-3}$	1.00

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
