# Peer review of "Error Analysis of a PFEM Based on the Euler Semi-Implicit Scheme for the Unsteady MHD Equations"

_entropy, 2022, doi:10.3390/e24101395_

Round 1

Reviewer 1 Report

The remarks are given in the enclosed file.

Author Response

请参阅附件。

Reviewer 2 Report

The manuscript corresponds to the subject of the journal Entropy and can be published after correcting the following comments:

1. It is necessary to bring the manuscript according to the instructions for authors (at the moment, the work is poorly perceived due to the large number of sections, with a poorly visible logical chain).

2. In the background, you need a more understandable statement of the goal and objectives of your research. It is also necessary to more specifically formulate conclusions on how you have achieved your goals and objectives of your research.

3. Why do we need a numerical example, what did the authors want to show in this section? In my understanding, the classic tests for the convergence and adequacy of research results are a comparison of the obtained data with already known solutions. It is also not clear how the numerical error data were obtained.

Round 2

Reviewer 1 Report

See the enclosed file.

Round 3
